# Building the Bridge to a Participatory Citizenship: Curricular Integration of Communal Environmental Issues in School Projects Supported by the Internet of Things

**DOI:** 10.3390/s23063070

**Published:** 2023-03-13

**Authors:** Manuel J. S. Santos, Vânia Carlos, António A. Moreira

**Affiliations:** CIDTFF—Research Centre on Didactics and Technology in the Education of Trainers, Department of Education and Psychology, University of Aveiro, 3810-193 Aveiro, Portugal

**Keywords:** transdisciplinarity, citizen kits, education kits, Internet of Things, participatory citizenship, environmental issues, school projects, citizen science, Domains of Curricular Autonomy, STEAM

## Abstract

Generally, there is much to praise about the rise in acknowledging the need for young citizens to exercise their rights and duties, but the belief remains that this is not yet entrenched in young citizens’ overall democratic involvement. A lack of citizenship and engagement in community issues was revealed by a recent study conducted by the authors in a secondary school from the outskirts of Aveiro, Portugal, during the 2019/2020 school year. Under the umbrella of a Design-Based Research methodological framework, citizen science strategies were implemented in the context of teaching, learning, and assessment, and at the service of the educational project of the target school, in a STEAM approach, and under Domains of Curricular Autonomy activities. The study’s findings suggest that to build the bridge for participatory citizenship, teachers should engage students in collecting and analyzing data regarding communal environmental issues in a Citizen Science approach supported by the Internet of Things. The new pedagogies addressing the lack of citizenship and engagement in community issues promoted students’ involvement at school and in the community, contributed to inform municipal education policies, and promoted dialogue and communication between local actors.

## 1. Introduction

Citizenship education traditionally aimed to prepare young people to take their place in adult society [1]. The values intended to be promoted are those of democracy, with a focus on human rights, a creative spirit, and an eager critical sense, creating effective opportunities that involve young citizens in solving their own problems. “Government of the people, by the people, for the people” remains the sovereign definition of democracy, as stated by Winston Churchill’s speech to the House of Commons in 1947.

The Portuguese Ministry of Education (ME) has recently published several documents regarding citizenship education, more specifically, Portuguese Citizenship Education Guidelines (PCEG) [2], National Education Strategy for Citizenship (NESC) [3], and Sustainability for Environmental Education Referential (SEER) [4]. PCEG states, “The practice of citizenship is a participatory process, both individual and collective, that calls for reflection and action on the problems experienced by each person and by society” [4] (p. 1), thus, defined to contribute towards the development of a just and equal society. The NESC aims to contribute to developing autonomous, responsible, and solidary citizens with others they interact with. It also seeks to lead young citizens to know and exercise their rights and social duties, always respecting others. SEER, on the other hand, kept in line with international policies to seriously address climate change, such as the 1972 Stockholm Conference; the 1975 Belgrade Conference; the 2000 United Nations (UN) Millennium Declaration; the 2016 UN 17 Sustainable Development Goals; The Paris Climate Conference (COP-21) of the UN Framework Convention on Climate Change; and recently, to reinforce the 1972 principles, 2022 Stockholm +50 UN Conference. The SEER is a guiding document for this theme implementation within the scope of Citizenship and Development (CD) subject that integrates the curriculum in the different cycles and levels of education and teaching in Portugal. The main objective is to prepare young students to exercise conscious citizenship, be informed about current environmental problems to effectively change attitudes and behaviors, and promote new environmental values. This objective is aligned with Stockholm + 50 Leadership Dialogue 1: reflecting on the urgent need for actions to achieve a healthy planet and prosperity of all, which stresses the need for promoting environmentally conscious syllabi and curricula [5]. Citizenship is the practice of someone playing an active role in their own community. For example, volunteer work, garbage-cleaning campaigns, local elderly care cooperatives, or involvement in decision-making about the municipal budget. In the same line of thought, it is current practice for students to exercise their right to choose where part of the school budget should be invested. Residents should become more involved in public life, so the role of government is to consider community initiatives, so-called “government participation”. A new way for citizens and government to work together is do-ocracy. Do-ocracy is a kind of organizational structure where citizens choose which roles they will play, as well as tasks, and carry them out; a do-it-yourself approach fosters a sense of responsibility towards a common goal.

Citizen Science (CS) is a growing phenomenon that echoes citizens’ contribution to generating scientific information, new knowledge, and understanding, a common name for a wide range of activities and practices that promote public awareness and involvement [6,7,8]. The design of CS activities in collaboration with citizens is a complimentary science approach to traditional sensor-based collection. CS engages citizens in data gathering activities, community-based environmental monitoring, data collection and interpretation, and providing answers to achieve full participation in science and policymaking. Citizens work together in collective decision-making to identify communal issues and find participatory solutions, which enables science to become more responsive to community needs. Therefore, communities can raise awareness and develop strategies and solutions to mitigate communal issues based on the data gathered. Citizens assume different roles with different levels of engagement, from a simple contributory level, through collaborative, or at a co-created level. In any of them, engagement can be top-down, led by the project leaders, or bottom-up, guided by the participants themselves [9]. The Internet of Things is a key enabler that provides a path to support data collection activities in a real-world CS project in a Smart City approach, using low-cost sensors for gathering and reporting data and cloud storage for hosting and storing data [10]. CS integration into school practice can strengthen students’ science and environmental literacy and promote their citizenship identity and active participation competence. Overall, the study aims to explore the educational value of CS strategies developed within an educational scope and the potential of the Internet of Things to contribute towards participatory citizenship, engaging citizens in communal issues, providing rich and detailed information, and proposing and contributing to co-created solutions [11]. Retaining citizens as volunteers is crucial to ensure their on-going long-term participation. So, encouraging teachers to integrate CS into their teaching practice will foster empirical evidence that citizen engagement can provide a local and contextual dimension to communal issues, being able to educate and engage communities. CS integration into school practice can strengthen students’ science and environmental literacy and promote their citizenship identity and active participation competence.

While there is generally much to praise about the rise in acknowledging the need for young citizens to exercise their rights and duties, the belief remains that this is not yet entrenched in young citizens’ overall democratic involvement. A lack of citizenship and engagement in community issues was revealed by a recent study conducted by the authors in a secondary school from the outskirts of Aveiro, Portugal, during the 2019/2020 school year, regarding students’ participatory citizenship attitudes [12]. The exercise of participatory citizenship should begin at an early age [13,14]; therefore, enhancing such an exercise in an educational context should be emphasized and an object of study.

The rest of the paper is organized as follows: Section 2 presents an overview of the related work. Section 3 describes the smart learning environment and the technical set-ups of the proposed systems. Section 4 describes the implementation of the designed systems, data interpretation, and discussion of evaluation results. Finally, Section 5 presents conclusions and future work that needs to be addressed.

## 2. Theoretical Framework

### 2.1. Citizenship Education

Launched in September 2012, under the United Nations’ umbrella, the Global Education First Initiative’s three main priorities are: (1) put every child in school; (2) improve the quality of learning; and (3) foster global citizenship. These three pillars are the core of the 2015 United Nations advocated model, supported by UNESCO, which are to be implemented until the 2030–Education 2030 Agenda and Framework [15]. The interconnected NESC and SEER are impacting the policy visions and practices of ministries of education around the world, which also impacts how teachers work but requires measures to achieve a Global Citizenship Education (GCE) [16]. According to UNESCO, GCE aims to enable students to assume active roles, both in their local community and globally, which will allow the construction of more peaceful, tolerant, inclusive, and safe societies [17] aligned with the objectives of Education for Sustainable Development (ESD) and the Sustainable Development Goals (SDG 4 on Education). UNESCO also points to the sense of belonging to a larger community and common humanity as global citizenship [18,19]. The World Conference on Education for All in Jomtien, Thailand, on 5–9 March 1990, may be considered the first global education initiative in the last decade of the 20th century. About 1500 delegates from 155 countries and representatives of some 150 governmental, non-governmental, and intergovernmental organizations, including the United Nations Educational, Scientific and Cultural Organization (UNESCO), United Nations Children’s Fund (UNICEF), United Nations Population Fund (UNFPA), United Nations Development Programme (UNDP), and the World Bank as key education stakeholders, attended the conference. Participants in the conference adopted the World Declaration on Education for All and a Framework for Action to Meeting Basic Learning Needs, highlighting the need to invest in education aimed at universalizing primary education. This declaration is in line with Millennium Development Goals [20] (replaced by the SDGs in 2015)—surely one of the most important global development initiatives [21]—that were to be achieved over the period 2000–2015.

Westheimer and Kahne [22] call attention to “participatory citizens” as those that take in hand the concept of the “good citizen”, and that can support democracy helpfully, pinpointing the need for educational programs to prepare students to engage in collective efforts. Such a citizen “actively participates in the civic affairs and the social life of the community at local, state and national level”, working together for the common good, with a strong collective faithfulness, to solve communal issues [22] (p. 240). Amagir et al. provide evidence that “the development of educational curricula that focus on the development of active citizenship” [23] (p. 57) should be considered as early as possible. Citizenship education could be the right host as it seeks to engage students to be more active, informed, and responsible [24]. Active citizen participation in a democratic society requires informed decision-making. Based on the effects of human activities on the environment, the Portuguese ME [4] highlights the need to enhance students’ knowledge that allows them to interpret and assess the communal surrounding reality so that they can formulate and debate argumentation that permits the sustainability of their positions and options.

### 2.2. The Internet of Things and Smart Cities

The Internet of Things (IoT) is a global network of “things” that are connected to the internet using several communication technologies [25] and can “communicate” with each other, aiming to monitor and control the physical world through collecting, processing, and analyzing data gathered by IoT devices, without human intervention [26]. IoT devices must be connected to the internet and use Transmission Control Protocol/Internet Protocol (TCP/IP), the internet network protocol, to communicate among themselves. Different communication protocols for data transmission in IoT and M2M systems can be used, such as IoT Data Communication Protocols; Message Queue Telemetry Transport (MQTT); Hypertext Transfer Protocol (HTTP), among others [27]. Accurate consumer-data provided by IoT sensors can be communicated to a broker by MQTT and be remotely accessed/controlled in real-time by IoT devices, computers, smartphones, or tablets [27]. MQTT requires low bandwidth consumption with efficient distribution of information—one or many receivers (subscribers)—with low power consumption; MQTT is ideal for M2M/IoT applications [28], with a tiny code footprint. There are several MQTT brokers and client libraries in the industry, such as Mosquitto, EMQ X, HiveMQ CE, HiveMQ, Thingstream, Adafruit IO, Eclipse M2Mqtt, IBM Watson IoT Platform Message Gateway, and many more. This article will not discuss their advantages, disadvantages, or functional differences. A more detailed comparison of some currently available MQTT implementations can be found in [27]. For the research purpose authors used the HiveMQ MQTT broker that provides a public MQTT broker (www.broker.mqtt-dashboard.com) (accessed on 2 January 2023). 

Still, there is no one-size-fits-all solution regarding communication technologies. One must consider throughput, range, and power consumption, etc. Several wireless communication technologies exist, such as short-range wireless, cellular, and low-power wide-area networks (LPWANs). Figure 1 presents a graphical correlation between the range and data rate of several IoT communication protocols.

Data gathered by sensors are sent to other IoT devices or a gateway and analyzed locally or sent to a cloud provider to be further analyzed. In the end, as Hosseinzadeh et al. [29] refer to, the main objective of IoT is to “connect anything to any object, at any time, and over any network/service”. Statista [30] forecasts an increase of three times the number of connected IoT devices from 9.7 billion in 2020 to more than 29 billion in 2030. IoT is assumed by Nižetić et al. as “one of the key pillars of the fourth industrial revolution” [31]. Over the last few years, the IoT has entered various sectors, so this network of sensors, actuators, and smart objects, whose purpose is to interconnect all things, has huge potential and usefulness benefits for the population [31]. IoT is used in many fields, such as agriculture, smart cities—including smart homes, smart grids, smart agriculture, smart lighting systems, smart transportation, smart healthcare, and augmented reality (AR) [32,33,34], alongside traffic, healthcare, security, and retail. Still, there are other fields, like education, where it is relatively scarce [31]. For instance, in the agriculture filed, IoT is often used in the process of crop nutrition monitoring, light, ph, temperature, and humidity, using sensors that send data from node to node to be processed and to make decisions or can change other devices automatically regarding system decisions [25,31,35,36,37]. 

The “employment of IoT in education is still considered at its early stages but is highly encouraged” [38]. As [39] state, the coming years will have a huge increase of connected devices, so this should be seen as an opportunity for educational institutions to incorporate IoT in the school environment [39,40,41], prepare students for the upcoming years’ challenges, making learning relevant to students. IoT in education provides learning concepts, such as e-learning, including media that transforms virtual classrooms into significantly more efficient places [42]. Based on the already overloaded school curricula, attention must be paid to incorporating IoT skills to bestow a plethora of skills and abilities to students. The ideal would be to have stand-alone courses to ensure enough time can be devoted to cover the various topics but could also be offered in line with CD subject context in a transdisciplinary effort with Information Communication Technology (ICT) or even at school projects/clubs.

High urbanization of cities and communities, alongside huge population increase, presents societal challenges such as climate change. Technology can be a “game change factor” in sustainable development, as well as a facilitator of citizens’ resilience [43], providing citizens with a sense of ownership of their surroundings [44]. There is a need to change, and the power “needs to be challenged” to an “inclusive city” that is “governed by powerful place-based democratic institutions” and create an equal society, instead of what is happening in an era of strong globalization where “place-less leaders” are gaining ground [45]. The aim is to advocate the need for a look at the collective power exercised by the citizen, an escape from the marketing approach, which treats people as consumers, towards a more democratic approach, which treats people as free and responsible citizens, with the right to be heard, a citizen-oriented approach, as stated by Hambleton [45]. 

As cities take advantage of emerging technologies, such as IoT, several facets of day-to-day urban life are undergoing enormous transformations [46]. The capitalization of technologies by smart cities paves the way for citizens to interact with cities’ physical infrastructure, services, and many more [40], transforming/fostering citizenship and enabling citizens’ right to the smart city. De Wall, Lange, and Bouw proposed The Hackable City research project as innovators regarding the “hackable city”, which put forward collaborative city-making approaches that empower stakeholders in an open and democratic community and can be “hacked” or appropriated by their citizens towards a common public good [44]. 

However, who has the power and the means to name an issue and put it on the agenda? Citizens may organize themselves in informal groups of neighborhood residents rather than waiting for new solutions to come into view or with local stakeholders around the issues they deem urgent. Three distinct sets of smart city visions were introduced by de Wall and Dignum: the control room; the creative city; and smart citizens—assuming that citizens should be linked with new roles and responsibilities, integrated into an ideological perspective that underlies the notion of citizen-centered smart cities [43,47,48]. The notion of “smart citizens” is opposed to the notion of the “control room” since it is not just the perspective of a collection of infrastructures and services in a top-down approach but rather a perspective of the community city, relying on citizens’ aspirations, problems, and skills, mobilizing themselves around common issues. Arnstein [49] defines three levels of typology of participatory citizenship: (1) citizen power; (2) tokenism; and (3) non-participation. Based on Arnstein’s work, Cardullo and Kitchin [50] and Cowley et al. [51] have explored different roles of involvement of citizens in smart cities, but both indicate that more active roles for citizens are not necessarily a desired goal. Indeed, what roles should be assigned to citizens—students are the focus of this paper—based on their individual choices and meeting the common good? In this paper, collaboration and co-creation in an open and sharing process (bottom-up approach) are assumed as the main role for students to assume, a citizen power form of participation in partnership with teachers, parents, local power, and local stakeholders. 

Almost all smart city infrastructures have sensors built into their core. Being easy to handle, this wireless network of low-cost and low power-consumption easy-to-use sensors covers a large area without any large space requirements, based on its wireless communication capability. Low-cost devices combined with advanced information systems can reduce the cost of daily life monitoring [52]. The enormous amount of data generated by the countless interconnected devices offer unique opportunities to solve urban challenges. The use of these sensors, combined with applications for smartphones, is making its path across domains such as noise, air quality, or radiation monitoring [53,54,55]. Air quality is a major concern in urbanized areas, so research on air quality is huge [56,57,58,59,60,61,62,63,64,65,66,67,68,69]. The European Union’s (EU) clean air policy is based on three pillars: ambient air quality standards, reducing air pollution emissions, and emissions standards for key sources of pollution. According to the European Environment Agency (EEA), air pollution is Europe’s biggest environmental health risk [70]. World Health Organization (WHO) [71] global air quality guidelines regarding particulate matter (PM2.5 and PM10), ozone, nitrogen dioxide, sulfur dioxide, and carbon monoxide states that millions of deaths and the loss of healthy years of life reflect exposure to air pollution. The report highlights that in 2020 in the EU, “96% of the urban population was exposed to levels of fine particulate matter above the health-based guideline level set by the WHO”. Setting pathways to cleaner air, water, and soil, the European Commission (EC) has placed air quality at the top of its priorities, which is why it is committed to continuing to improve air quality in strict alignment with EU air quality standards and WHO recommendations. Due to human activities causing polluting emissions, Europe’s air quality has deteriorated considerably. To fight climate change, environmental pollution, biodiversity loss, and unsustainable use of natural resources, the EC stated in 2021 the zero-pollution vision for 2050 under the European Green Deal’s Zero Pollution Action Plan [72].

The second most damaging environmental threat to human health in the modern day is noise pollution. WHO published noise guidelines to provide recommendations for protecting human health from exposure to environmental noise [73]. According to the report, in the EU, “at least 100 million people are affected by road traffic noise and in western Europe alone at least 1.6 million healthy years of life are lost as a result of road traffic noise”. Regarding road traffic noise, the WHO recommended reducing noise levels produced by road traffic below 53 decibels (dB) and below 45 dB for night noise exposure. Recommendations regarding railway, aircraft, wind turbine, and leisure noise are also addressed. In a recent survey carried out by Kaźmierczak et al. [74] in 2021 in Polish cities on 770 residents’ opinions on their perception of the city’s acoustic environment, about 85% of respondents stated that city sounds matter to them. A Google form was sent out to a non-probability sample city resident via the internet. The authors used an extensive snowball sampling method to reach city residents. Other studies have been conducted aiming at smart cities’ noise-related issues [75,76,77,78,79,80,81].

The increase in population in urban centers entails a huge increase in waste production, which has certainly become a significant challenge. Urban waste and recycling management include, among other actions, maintaining a healthy and clean city, production, transport, collection, storage, sorting, treatment, recovery, and disposal, so the collaboration of citizens is fundamental [82,83,84,85,86,87,88,89,90,91,92,93,94]. The World Bank [82] predicted that by 2050 the annual solid waste generation would reach around 3.40 billion tones. The report refers to the need to make citizens environmentally aware of reducing the amount of waste produced, separating it, and placing it in appropriate containers. There is a close link between the success of sustained solid waste management and public engagement. Of the countries and cities studied in Kaza et al. [82], several make waste management information available to the public. The most common types of available information include collection schedules and waste drop-off locations, budgets and fees, local statistics on waste generation and composition, and community programs and recycling campaigns. Between Christmas and the end of 2022, waste collection in Portugal recorded high values, with residents of the Lisbon Metropolitan Area producing more than 21,000 tons and those in Porto exceeding 10,000 tons [95]. City councils define sweeping plans for the city, with frequencies that vary in garbage collection and recycling according to the characteristics of each zone and the time of year. City councils are often faced with unnecessary trips as some of the containers are empty or with a load of less than 50%, thus being faced with an inefficient process, a great environmental impact, and high costs for cities. There is a need for intelligent technology to support the municipal solid waste management system. 

Ali et al. [85] identified solid waste management as critical in their study regarding IoT systems smart waste bin monitoring for smart cities. Several comparison tables are provided regarding existing proposed systems in the market. From the analyzed systems, features such as cloud storage, real-time data, waste prediction, GIS, and bin locations are provided. Most systems are grounded at LoRa and GPS/GSM communication, providing bin locations and allowing cloud storage. All systems are Arduino-based microcontrollers and rely mostly on Java and C++ software. Only one used the ThingSpeak platform and Google Map API. In Mozambique, the MOPA (https://www.mopa.co.mz) (accessed on 4 January 2023) project engages citizens in waste management problems to report through a digital platform. Relaying these problems through an open-source map, the city council will recruit micro-enterprises for waste collection. There are already alternatives in the Portuguese market that make it possible to take advantage of the Internet of Things and sensors to carry out integrated waste management (http://www.360waste.pt/indexpt) (accessed on 4 January 2023). Garbage container real-time data is certainly an added value for the city and citizens, as one can take advantage of the information gathered by sensors to determine which garbage container is full and cannot be avoided. 

Even though some proposed system enables citizens to report collected waste, there is a need to engage citizens from the early stage of the definition of the system, i.e., in a bottom-up approach [96]. Citizens are merely collecting data (citizen-sense approach) but not engaging from the early beginning regarding issue definition, and in the last phase regarding analysis and strategies definition to mitigate the issues, a citizen-science approach. Through citizens’ participatory endeavors regarding waste and recycling, municipal management strategies can be designed more effectively, so long-term outcomes of such strategies would be achieved.

### 2.3. Educational Project of the Target School

Aiming for educational success and the formation of upstanding citizens with full rights and duties, four major Areas of Intervention in the target school’s 2022–2025 Educational Project (EP) [97] are selected. The EP is an instrument that aims to bring together the educational community, focusing on improving students’ results, training for citizenship, and, lastly, the school’s self-evaluation process. With a vision focused on the education of citizens of a world in fast transformation, the EP aims to emphasize the work that has been carried out in the pursuit of equal opportunities, opening the doors to the local community, which allows the educational community to be involved in active participation, thus fostering a sense of belonging. In line with the NESC, the school highlights the importance of reinforcing the paradigm of partnerships with entities outside the school, as well as the articulation with local authorities and municipalities, fostering the creation of (1) collaborative and networked practices and work with an emphasis on sharing reference practices in Education for Citizenship; (2) networking between schools; and (3) networking between schools and stakeholders. To achieve such objectives, and for each main area, general objectives and the respective action strategies and measurement indicators were defined to monitor and evaluate the results. Goal 2 of the “Assessment, Teaching and Learning” intervention focuses on “Enhancing citizenship practices”. To achieve this, seven action strategies and respective measurement indicators were defined (see Table 1).

As referenced in NESC, the successful implementation of this strategy is strongly linked to the opportunities that schools give students to get involved in decision-making, specifically in those that affect them. So, regarding Goal 9, “Create opportunities for student participation in decisions about their education”, subordinated to the intervention area “Organization Culture”, five action strategies, and respective measurement indicators were defined (see Table 2). 

Regarding Goal 11, “Adopt behaviors that contribute to the sustainability of the planet and the common good”, under the umbrella of the intervention area “Sustainability, health and well-being”, in line with SEER Sub-theme C: Ocean literacy (Understand the importance of the oceans for the sustainability of the planet), three action strategies and respective measurement indicators were defined (see Table 3). 

### 2.4. Study Aims

Gaps were detected regarding students’ participatory citizenship, so main objectives were drawn: (1) implement CS strategies in the context of teaching, learning, and assessment, and at the service of the educational project of the target school; (2) train the educational community in the use of collective conscience digital platforms. The study’s specific objectives are: (1) to promote students’ involvement at school and in the community; (2) to promote the dialogue and communication between local actors; (3) to contribute to informing municipal education policies (see Table 4). 

## 3. Materials and Methods

This section presents the research milestones and the description of the smart learning environment as well as the design and implementation of CS strategies at Smart School Lab.

### 3.1. Research Milestones

The investigation began in the 2018/2019 school year with the creation of the smart educational community. The necessary steps for its creation were the following: (1) state-of-the-art literature review; (2) document analysis; (3) geomentors mobilization; (4) local needs and contextual purposes regarding School and Municipality; (5) roles and commitments. The creation of the community started with the first civic co-creation workshop—“Smart Schools: civic co-creation of smart educational communities”, where the project’s objectives were discussed, and local issues were raised. These questions formed the basis for the next phase of the project. To participate in this workshop, 16 teachers from the target school of the study from different disciplinary areas signed up (see Figure 2).

Design-Based Research (DBR) was the adopted methodological framework. The first cycle was carried out in the 2019–2020 school year aiming at the design and implementation of CS strategies. To reach this milestone, the necessary steps were: (1) definition of priorities for action; (2) promotion of training sessions and workshops of civic co-creation to develop strategies regarding georeferenced data gathering and participatory analysis, aiming at curricular integration (see Table 5). 

To improve stakeholder engagement (city leaders and citizens), training sessions and workshops were attended by teachers, the school board, the parents’ association, the parish council president, and the municipality (see Figure 2). 

A 50-h teacher training course was also conducted in a co-creation process with a smart educational community and stakeholders. A group of 18 teachers that taught grades 7–12 and encompassed several disciplinary areas attended the teachers’ training course. Research has occurred in six face-to-face sessions and six online sessions. More information regarding phase 1, conducted in 2019, is available in Santos et al. [98].

A second DBR cycle was implemented in the 2021/2022 school year, with a refinement of prior CS strategies designed, implemented, and analyzed in a co-creation process in the first cycle. An initial three-hour workshop was carried out with four teacher participants of the study in the early school year, aiming for the representation of expertise within the STEAM field was considered. Each teacher majored in one of the following: Physics and Chemistry, Mathematics, Arts, and Technology and Engineering. Researchers presented seven didactic kits co-developed in a teacher training course in 2020, and their functionalities were discussed, as well as their pros and cons regarding the implementation or adaptation by the teachers.

### 3.2. Smart Learning at Smart School Lab

The Smart School Lab (SSL) is a training and learning space where schools and municipalities learn collaboratively through citizenship strategies. Implemented in 2018 in the target school of the study, several workshops and training sessions have been promoted, aiming to create an intelligent educational community. In a smart city topology, all educational system’s local actors work together to co-create CS strategies with participatory data collection gathered by sensors—real-world data collection—microcontrollers and IoT, discovering and building apps, and engaging in solving important local issues. Smart environments automate field data collection in the natural environment, like air quality, noise, and waste management. The Smart School Lab logo (see Figure 3) incorporates elements such as air and soil quality, as well as cycling infrastructure, reflecting its focus on smart city problematics.

Based on Hambleton’s triangle of engaged scholarship [45], SSL focuses on research with education and the third pillar, “policy and practice”. Intrinsically linked to the University of Aveiro (UA), SSL takes advantage of research carried out by the UA, involving researchers and practitioners in a co-creation process. Locally elected leaders tend to have a more holistic understanding of the challenges their communities face [45]. To achieve SSL smart environment, techniques that allow new participatory models based on effective changes in community behavior must be fostered [98]. This smart environment is grounded on an assembly of sensors, microcontrollers, computers, displays, actuators, and other computational elements that control and monitor communal environmental issues, such as pollution, greenhouse gases, waste, safety routes, etc. Aiming to achieve a smart learning environment, teaching, and learning approach at SSL flawlessly incorporates IoT technology and CS strategies. Table 6 lists some of the properties of the environment that need to be captured and how they can be measured.

In recent years, many European countries have been revising their compulsory curricula to introduce basic concepts associated with Computer Science, aiming to develop students’ Computational Thinking (CT) skills [100]. Recently the European Commission has introduced a new Digital Education Action Plan (DEAP) 2021–2027 [101] designed to tackle emerging challenges to delivering education. The two DEAP priority areas and actions are: (1) fostering the development of a high-performing digital education ecosystem and (2) enhancing digital skills and competencies for digital transformation. Each priority area has developed a series of actions to achieve each goal. DEAP priority area number 1 has six series of actions, number 6 being “Implement Artificial intelligence and data usage in education and training”. Related to the use of extended reality (XR) in education and training, this action focusses on developing a basic knowledge of Artificial Intelligence to exploit its potential by citizens. Action 10, on the DEAP priority area number 2, proposes a “Council recommendation on improving the provision of digital skills in education and training”. The major outputs will be: (1) improve the provision of digital skills in education and training; (2) ensure that 65% of Europeans have at least basic digital skills by 2025. To achieve “Digital Opportunity Traineeships”, action 12 ensures that vocational education and training (VET) has “the opportunity to gain hands-on professional experience in digital fields demanded by the labour market”. In Portugal, about 40% of learners in secondary education undertake a VET program, which leads to a double certification [102]. A great majority of students who attended the SSL club come from VET. These students are very receptive to project-based learning. In 2012 the European Schoolnet (http://www.eun.org/) (accessed on 3 January 2023) created the Future Classroom Lab (FCL) (https://fcl.eun.org/) (accessed on 3 January 2023) focused on rethinking pedagogy, technology, and design in Brussels classrooms, supporting the dissemination and expansion of innovative pedagogies aided by Information and Communication Technologies. These spaces privilege the student’s action, which fosters motivation, creativity, and involvement in the individual and/or collective construction of knowledge. SSL goals and strategies align with DEAP, aiming to foster students’ and teachers’ digital skills and competencies regarding digital transformation, delivering 21st-century learning skills.

### 3.3. Design of Citizen Science Strategies at Smart School Lab

The CS strategies designed in the Smart School Lab are divided into two main goals: design strategies for citizens’ engagement in collecting data; design strategies for sensors collecting data. Table 7 presents the strategies used to achieve these goals, the number of users, and the answers to the Goal. Users’ roles were assigned in the ArcGIS platform according to each profile: student, teacher, stakeholder, and administrator. By category, the registered users are 203 students, 26 teachers, 2 stakeholders, and 4 administrators.

#### 3.3.1. Citizens’ Engagement in Data Collection

I—Proposed systems architectures

The first proposed system architecture is divided into four main components: positioning technology, Mobile device, form-centric data gathering solution, and server-side. These four components aggregate the system tasks’ information, GPS position, and users’ result information (see Figure 4).

Positioning technology: the system supports GPS communication, integrating a GPS NEO-6M module with ceramic antenna, signal super with data backup battery and default Baud Rate 9600 bps.

Mobile device: the system should be running on a GPS-enabled mobile device, either a smartphone or tablet. The equipment must be connected to the Access Point “Bike Track” to be accessed by a browser.

Server side: the web server stores and manages the information sent by the user via mobile device browser, including problems detected by the citizen and their GPS coordinates. To be used in outdoor spaces, the proposed Web server needs a LoRa 868 Mhz connection to send the data to the FTP server, thus updating the citizens’ reported problems. The FTP server stores the data sent by the Web server for later upload to Esri Survey123^®^ (https://survey123.arcgis.com/) (accessed on 5 January 2023) application.

Form-centric data gathering: the data loaded into the geographic information systems (GIS) Survey123 application smart form, a new data collection method that collects data and geospatial information, allows the parish council to analyze the data in real time to support decision making. Based on a user-friendly GIS-based map created in real-time, citizens can see their contribution instantaneously as data is uploaded from the field. 

The second proposed system architecture is divided into three main components: positioning technology, mobile device, and microcontroller. These three components aggregate the system tasks’ information, GPS position, and users’ result information (see Figure 5).

Positioning technology: the system requires GPS mobile communication and Bluetooth to be turned on.

Mobile device: the system should be running on a GPS-enabled mobile device, either a smartphone or tablet: Bluetooth turned on is mandatory.

Microcontroller: an ESP32 microcontroller with an internet connection and Bluetooth connectivity installed in the study’s target school will receive the data collected by citizens via Bluetooth. These data are published in an MQTT broker for later subscription by another microcontroller installed in the parish council. As an aggregator, the microcontroller installed in the school allows citizens to receive the data collected by other users via Bluetooth.

Third through sixth proposed systems architectures are divided into three main components: positioning technology, mobile device, and form-centric data-gathering solution. These three components aggregate the systems tasks’ information, GPS position, and users’ result information (see Figure 6).

Positioning technology: the system requires GPS mobile communication to be turned on.

Mobile device: the system should be running on a GPS-enabled mobile device, either a smartphone or tablet.

Form-centric data gathering: the data loaded into the GIS Survey123 application smart form allows the Municipality to analyze the data to support decision-making. GIS-based maps created in real-time as data is uploaded from the field allow citizens to see their contribution instantaneously.

II—Data collection

The first proposed system collected data in the 2018/2019 school year on issues related to garbage, recycling, and bicycle paths/safety routes (see Figure 7). 

The second proposed system collected data in the 2021/2022 school year on issues related to recycling, urban problems, and other non-categorized issues. Data collected by citizens include the detected issue and its geographic location using GPS technologies. Citizens could see “Parish council work” regarding reported issues in their neighborhoods, increasing transparency on how and what was done and when issues were solved.

The third proposed system collected data in the 2018/2019 school year on issues related to marine debris. Data collected by citizens include their names and photo and their geographic location using GPS technologies. Based on the georeferenced trash collection at Praia da Costa Nova, Portugal, the students produced a georeferenced digital narrative using Esri StoryMaps (https://storymaps.arcgis.com/) (accessed on 2 January 2023). Data gathered was provided to the municipality in Coastwatch Europe (https://coastwatch.org/europe/) (accessed on 2 January 2023) local seminar and at the “16th Esri User Conference”. Coastwatch Europe is an environmental network of groups aiming to protect wetlands and raise awareness on European coast local communities regarding the protection and sustainable use of coastal resources, engaging citizens in environmental planning and management. Aiming to raise students’ awareness of responsible, active, and participatory citizenship, with ecological behavior that allows valuing the environmental heritage, the same application was used to collect and georeference the produced waste in the target school of the study (see Figure 8).

The fourth proposed system collected data in the 2019/2020 school year on issues related to bike paths/safety routes. The essential learning objectives were: (1) prepare students to be democratic, participatory and humanist citizens (safety road); (2) negotiating the solution of conflicts promoting ecological sustainability, being interventionist, taking the initiative and being entrepreneurial; (3) discussing points of view, analyze documents, collect data, make syntheses, formulae hypotheses, observe experiences, learn to consult and interpret different sources of information, answer questions, formulate other questions, evaluate situations, outline solutions to problems, express ideas orally and/or in writing; (4) develop the ability to select, analyze, critically evaluate information in concrete situations; (5) develop group work skills: confrontation of ideas, clarification of points of view, argumentation and counter-argumentation in solving tasks, with a view to presenting a final product; (6) develop abilities to communicate ideas orally and in writing; (7) be critical and present well-grounded positions regarding the defense and improvement of quality of life. Data collected by citizens include the name and photo of the issue and its geographic location using GPS technologies (see Figure 8). Based on the georeferenced issues, the students produced a digital narrative using Esri StoryMaps. Based on this work, a safe route for cyclists and pedestrians is currently being built at the municipality’s request.

The fifth proposed system collected data regarding municipality toponomy in the 2019/2020 school year. Students tend not to know the underlying history of many streets in the municipality, intensified by the factor that parents do not transmit this legacy of information about the streets and culture of the municipality to their children. The municipal project made it possible to intensively collect information on the toponyms of the streets, alleys, gardens, and avenues of the municipality of the city under study. An active learning environment was created under the theme of local toponymy to develop students’ knowledge and collaborative abilities and to allow them to learn in a dynamic and participatory way. Students who participated in data collection with the Survey123 application designed georeferenced routes based on toponymy (see Figure 8). The project constituted an excellent challenge for knowledge and dissemination of the land, a way of valuing the place where one was born and where one lives, developing a sense of belonging and, simultaneously, identity and civic awareness. There is evidence of the increase and strengthening of the culture and identity of the municipality by the students, thus valuing its identity and roots. Since these strategies have sharpened the students’ curiosity about the streets where they live, it is imperative to continue with activities of this type so that the historical legacy of the streets is not lost.

The sixth proposed system collected data in the 2019/2020 school year regarding charity (human rights) and is related to the exponential increase in the number of requests for help from residents of the target local community related to the COVID pandemic breakout. The charity mission is anchored in integral human development, as well as in defense of the common good, promoting the sharing of goods and assistance in calamity and emergency situations. This project aims at young people to be able to: (1) analyze and critically question reality, evaluate, and select information, formulate hypotheses, and make informed decisions in their daily lives; (2) be free, autonomous, responsible, and aware of themselves and the world around them; (3) dealing with change and uncertainty in a rapidly changing world; (4) continue learning throughout life, as a decisive factor in their personal development and social intervention; (5) know and respect the fundamental principles of a democratic society and the rights, guarantees, and freedoms on which it is based; (6) value respect for human dignity, for the exercise of full citizenship, for solidarity with others, for cultural diversity; and (7) reject all forms of discrimination and social exclusion. In a more simplistic way, it is intended, then, that students: (1) aspire to achieve a job well done, with rigor and overcoming challenges; (2) persevere in the face of difficulties, be aware of oneself and others; (3) have sensitivity; and (4) be sympathetic to others. A GIS-based map with information about people who needed help from Caritas in the target city of the study was drawn based on a georeferenced form built by the students. A georeferenced form was used with the Survey123 application, which allowed, confidentially, to know where the families in need of help lived and what kind of help was essential to provide. 

#### 3.3.2. Sensors Data Collection

I—Proposed Systems Architecture

The first proposed system architecture (SmartWaste) is divided into three main components: food waste management, microcontroller, and projection. These three components aggregate the systems tasks’ information and users’ result information (see Figure 9).

Food waste management: the system takes advantage of a 20 kg aluminum load cell and an HX711 converter module with 24 high precision A/D converter chips requiring 2.6–5.5 V supply voltage, connected to an ESP32 microcontroller.

Microcontroller: an ESP32 microcontroller with an internet connection installed in the kitchen of the target school of the study will receive the data collected by the food waste management system. The data is published to an MQTT broker for later subscription by a computer/laptop/tablet installed in the school canteen. 

Projection: the computer/laptop/tablet installed in the school canteen subscribes to MQTT Broker, and displays received information so students can see food waste in real-time.

The second proposed system architecture (SmartAir) is divided into five main components: CO_2_ sensor or PM sensor, microcontroller, cloud server, client, and traffic light microcontroller. These five components aggregate the systems tasks’ information, cloud storage, and users’ result information (see Figure 10).

CO_2_ sensor/PM sensor: the proposed system collects data on two different but complementary air pollutants, CO_2_ and particulate matter (PM2.5 and PM10). To gather CO_2_ data, the system takes advantage of an Infrared Carbon Dioxide Sensor connected to an ESP32 microcontroller, with an effective measuring range from 400–5000 ppm. This sensor is based on non-dispersive infrared technology with good selectivity and oxygen-free dependency. The sensor’s main characteristics are high sensitivity, high resolution, low-power consumption, fast response, anti-water vapor interference, no poisoning, and high stability. Regarding PM dust concentrations, a Plantower PMS5003 (https://www.plantower.com/en/products_33/74.html) (accessed on 21 December 2022) low-power consumption (below 100 mA on working mode and below 200 μA on standby mode) laser dust sensor is integrated into the proposed system. The sensor uses the laser light scattering principle to measure the value of dust particles suspended in the air with a sensitivity of 50%-0.3 μm and 98%-0.5 μm and larger and a 1 μg/m³ resolution.

Microcontroller: an ESP32 microcontroller with an internet connection will receive the data collected by the CO_2_ sensor and display a color-coded light according to the returned value (visual information using a traffic light analogy) and its value on a 0.95-inch OLED display.

Cloud server: the data is published to an MQTT broker and the ThingSpeak IoT Cloud Platform channel for later subscription by a computer/laptop/tablet installed.

Client: a computer/laptop/tablet displays graphic information in real-time on a browser or a CVS file.

Traffic light microcontroller: the subscriber microcontroller gets de data provided by the MQTT Broker and displays the same visual information provided by the first microcontroller.

The third proposed system architecture (SmartSound) is divided into four main components: sound sensor, microcontroller, cloud server, and client. These four components aggregate the systems tasks’ information, cloud storage, and users’ result information (see Figure 11).

Sound sensor: the proposed system collects data from the surrounding environment regarding sound level. To gather sound data, the system takes advantage of a Gravity Sound Level Meter V 1.0 Sensor, connected to an ESP32 microcontroller, with an effective measuring range from 30 dBA to ~130 dBA and a measurement error of ±1.5 dB. The sound level meter is also known as a decibel meter or noise meter and is basically a noise measurement instrument. There are three types of frequency weightings, and it consists of A-, C-, and Z-weighting. The electronic filter integrated inside the sound level meter correlates the objective measurements with the human subjective response—frequencies between 500 Hz and 6 kHz. The sensor is factory calibrated with the A type, the most used of a family of curves defined in the International standard IEC 61672:2003.

Microcontroller: an ESP32 microcontroller with an internet connection will receive the data collected by the sound sensor and display in real-time its value on a 0.95-inch OLED display, the maximum and minimum recorded and textual information indicating the status: restful, uncomfortable, dangerous.

The fourth proposed system architecture (SmartGreen) is divided into seven main components: sensors; microcontroller (connected to the sensors); cloud server; microcontroller; mobile device; water storage tank; sprinkler. These seven components aggregate the systems tasks’ information, cloud storage, and users’ result information (see Figure 12).

Sensors: the proposed system collects data from the surrounding environment regarding capacitive soil moisture level, air temperature, and humidity. To collect the data, the system uses a DHT11 Temperature and Humidity sensor and a soil Photometric Electrolyte Sensor built-in an ESP32 LilyGo^®^ TTGO T-Higrow. 

Microcontroller (connected to the sensors): an ESP32 microcontroller with Wi-Fi and Bluetooth communications sends the data collected by the sensors to an MQTT Broker and to FireBase Realtime Database. When requested, the data is sent via Bluetooth to the dashboard of an APP built by the students.

Cloud server: the data is published to a FireBase Realtime Database with MQTT for later subscription by a microcontroller.

Microcontroller: an ESP32 microcontroller with Wi-Fi and Bluetooth communications gets the data collected by the sensors by an MQTT Broker and to FireBase Realtime Database. The processed data will turn on/off a solenoid allowing water to reach the sprinklers, depending on the real-time data sent to the microcontroller by several water level sensors mounted on the water storage tank. When requested, the data is sent via Bluetooth to the dashboard of an APP built by the students.

Water storage tank: collect the rainwater for greenhouse irrigation. With a couple of liquid level water sensors, measurement of the liquid level in the fixed container, whether it is too high or too low, is updated in real-time to the microcontroller. With a capacitive liquid level sensing the induction principle, the sensors have an induction thickness range of up to 20 mm and a level error of ±1.5 mm. Strong compatibility through a variety of non-metallic containers, such as plastic, glass, ceramics, etc., will be used in a plastic water storage tank. A 12 V 10 Watts/0.50 Amps ¼″ solenoid valve with a pressure range of 115 psi and Cv 4.8 (approximately 36 GPM @ 60 PSI) flow rate will be used as flux control to activate de sprinklers.

Mobile device: in the system, the SmartGreen dashboard APP should be installed. The device should be running on a Bluetooth-enabled mobile device with Internet access: either a smartphone or a tablet.

II—Data collection

In line with the European Commission [103] measures to protect the environment and human health by preventing or reducing the impacts of waste generation, aiming principles of the sustainable development goals, SmartWaste will attempt to prevent food waste at school. United Nations SDG 12 (https://sdgs.un.org/goals/goal12) (accessed on 18 December 2022) states that 17% of total food is wasted at the consumer level. This system is being developed during the 2022/2023 school year and is expected to be fully functional by the time of writing this manuscript, and aims to: (1) foster organized and responsible civic participation in the definition and management of the common good; (2) innovate in entrepreneurship with community support services, which appeal to sustainability and problem-solving, and society’s needs; (3) promote sharing and disseminating knowledge about the economic/environmental effect; (4) promote active and responsible civic participation; (5) sensitize the educational community to healthy eating habits (type of food/products); (6) change behaviors based on the results obtained. The food weight (FW) will be measured in grams by an electronic kitchen scale designed and developed by the students with teachers’ supervision, with a capacity of 20 kg. Direct measurements will be taken each day in two stages: the amount of food served and the amount of food wasted using the following formula: $$\mathrm{FW}\;\left(\%\right)=\frac{\mathrm{Food}\;\mathrm{Waste}}{\mathrm{Food}\;\mathrm{Server}}\times 100$$

SmartAir was developed in the 2020/2021 school year, and its main objective was to sensitize students and their families to use soft mobility once the number of cars in the area near the target school was a reality. There is evidence regarding the decrease in zinc and iron in food associated with rising atmospheric carbon dioxide levels [104,105,106]. With the increase in the number of students coming to school by car, it was decided to study possible changes in CO_2_ concentration during the most critical phases of the day, i.e., the first and last hours in the morning and the afternoon. Installed at the entrance to the school, the collected data changed little during the day. However, this triggered other initiatives developed for and by the educational community to raise awareness about the need to change behaviors regarding mobility. SmartAir, focused on promoting sustainable mobility, was the winner of the 2021 edition of the Energy Up School Award, promoted under the Galp Foundation’s Future Up educational program, which consists of installing solar panels up to a value of 20,000 euros (see Figure 13). 

Still focusing on soft mobility, several awareness sessions were held in the surrounding area of the target school (see Figure 14) for the use of this type of mobility, with the active participation of the parents’ association, the president of the parish council, and the students.

To overcome this problem, a new version of SmartAir was developed in the 2021/2022 school year to gather PM polluters. Suspended particles with a maximum diameter of 2.5 μm are considered the most harmful and dangerous pollution to human health. Due to its fine nature and high capability to penetrate directly into the bloodstream, several problems underlie this pollutant. From exacerbated asthma arrhythmia, among others, there is a strong relationship between the intensification of symptoms of diseases related to the circulatory and respiratory systems and the pollutant. PM10 is considered harmful and has a negative effect on the respiratory system because it contains benzopyrenes, furans, dioxins, and carcinogenic heavy metals. During the COVID-19 pandemic, the rooms had to be continuously aired, according to the Portuguese Directorate-General for Health (DGS). In a statement, the DGS stated the need to airing spaces in schools between classes. The DGS guidelines [107] regarding the opening of the 2020/2021 school year state that schools must keep windows and/or doors open to allow better air circulation. Monitoring the air in the classrooms was, therefore, a necessity. The system was reconverted for indoor CO_2_ collection (SmartInAir). The data collected in two classrooms at the school made it possible to measure changes in CO_2_ concentration during class periods and breaks. CO_2_ and PM data collected is available in open access on the school’s website and in CSV format on the ThingSpeak platform.

SmartSound was developed in the 2021/2022 school year, and its main objective was to collect data in different places inside and outside the school, with curricular integration of STEAM activities. The data collected is available in open access on the school’s website and in CSV format on the ThingSpeak platform.

SmartGreen is under development in the 2022/2023 school year and aims to build a smart irrigation water management system supported by IoT technologies and sensors. The use of wireless sensor network technology “has the potential to increase the efficiency of irrigation water management” [108]. The data collected by the sensors will be forwarded to Firebase using an MQTT broker. Data will be stored in the Realtime Database. Published data will be subscribed by a microcontroller. The notification contains information about the plant’s nutrition, mainly moisture level. If not in accordance with the conditions for plant growth, and water is in need, then a solenoid valve will be turned on, which triggers the irrigation pump to restore the plant’s soil moisture stability. When there is enough moisture, a command is sent to deactivate the solenoid. This process is dependent on the existence of water in the water storage tank. A couple of capacitive sensors will send irrigation control data to the microcontroller. A SmartGreen dashboard APP should be installed on a Bluetooth device with internet access, which will allow viewing sensors’ status and remotely activating the smart irrigation system.

## 4. Results and Discussion

This section describes the design and procedure of the IoT-based CS co-created strategies and their curricular integration and presents the results. IoT wireless network-based sensors with a limited and unlimited power supply connected to a network are also considered.

### 4.1. Curricular Integration of Communal Environmental Issues in School Projects

As a result of increasing globalization and technological development, and in line with the Ministry of Education and Science Decree-Law N°. 55/2018 [109], it is essential that students develop skills for integrating emerging knowledge and solving complex problems. These skills presuppose the implementation embodied in Decree-Law N°. 54/2018 [110], in an inclusive way, of meaningful learning for all students based on each student’s specific contexts and needs. In this context, the possibility arose for each school to carry out an autonomous and flexible management of the curriculum, an instrument that must consider local realities. Management must be coordinated between all those involved in the educational process, in a close dialogue between teachers, students, families, and the local community focused on building knowledge that allows all students to achieve the skills set out in the students’ profiles at the end of compulsory education [111]. The Domains of Curricular Autonomy (DAC) thus constitute a curricular option for interdisciplinary work and/or curricular articulation, whose planning should preferably involve all disciplines, which will allow the mobilization of knowledge from different disciplines/disciplinary areas. It is intended to develop interdisciplinary and/or curricular articulation work that allows valuing the “transdisciplinarity of learning, mobilizing different literacies (…), promoting scientific knowledge, intellectual curiosity, critical and interventional spirit, creativity and collaborative work” [109].

The curricular integration of CS strategies took place in two Design-Based Research (DBR) cycles. The first moment was embodied in the DAC moments, and the second cycle was implemented in a Science, Technology, Engineering, Arts, and Mathematics (STEAM) approach, with the refinement of prior CS strategies designed, implemented, and analyzed in a co-creation process in the first DBR cycle. The integration of STEAM, a multi-discipline approach to teaching that enhances students’ digital skills and competences, fostered by IoT technologies, ensures the interdisciplinary dialogue between the disciplines, with new learning scenarios that promote students’ participation in community issues [39,41,112,113].

#### 4.1.1. Citizen Science Strategies Design

Table 8 and Table 9 present an overview of the projects listed in Section 3 of this study about Citizens’ engagement in data collection and the use of sensors to collect data.

#### 4.1.2. Citizen Science Procedures—Proposed Systems

The Waste/garbage, Bike paths/safety routes, Toponymy, and Charity projects initially require users to install the Survey123 application on their mobile phones. After authentication, surveys already downloaded and available for use will appear on the Survey123 dashboard. To proceed with the process, the user must access the settings and download the desired surveys (see Figure 15). All surveys have the same online interface for managing and analyzing the collected data.

Once the survey has been downloaded, the user can start collecting data. Collection can be done with or without Internet access. If the user does not have internet access available, the data collected is stored on the mobile phone for later synchronization by the user, proceeding with the publication of the data to the Survey123 platform.

Waste/garbage survey setup: using the proposed system, the survey consists of two activities: “Garbage hunting in the school yard” and “Monitoring sea waste”. The user must select the intended activity, write the date, year, and class, and select the type of garbage the user intends to reference. Different types of waste must not be grouped together, nor must several of the same category be joined. If location is activated on the mobile phone, the user can take advantage of the GPS location; if the user has not activated GPS location, navigation through the map is provided. Finally, the user should take a photo of the garbage in the place where it was found and collect it. (see Figure 16). 

The application provides a map with a detailed view of the collected data. (see Figure 17).

Toponymy survey setup: using the proposed system, the created survey consists of collecting information about the streets where the students live. Once the application is opened, the user should select the survey and start the process by indicating the name of the town where s/he lives and then writing the name of the street. In the following steps, the day and time at which the data is being collected are recorded, with a photographic record of the name of the street and some point of interest that the user located on the map. It is intended that the user indicates the levels of interest in the identified point: social, educational, sports, leisure, health, gastronomic, or any other that has not been previously defined. In the last field, the user is asked to briefly describe some relevant aspects of the identified point of interest, such as the date of construction/opening, the function, personalities associated with its creation, currently responsible for preservation or management, or other aspects.

Charity survey setup: using the proposed system, the created survey consists of collecting information about Human Rights in cooperation with Cáritas in the target city of the study, aiming to help this institution during the COVID-19 pandemic. Application users will be citizens who need the support of the institution. These must indicate their personal information and that of their household, including the location on the map, as well as the food and products needed. The information is too sensitive and cannot be presented in this study. The institution invited the students who participated in the construction of the survey to support the distribution of food once a month. Having started the project in 2020, it was only in September 2022 that the students stopped supporting the institution because they had completed secondary education, continued their studies at university, and studied in another city.

Bike paths/safety routes/recycle/waste setup: one proposed system collects information about safety routes, recycling, and waste management. The equipment comprises a three-dimensional structure with a LILYGO^®^ TTGO T-Beam V1.1 microcontroller ESP32 868 Mhz with Wi-Fi, Bluetooth, and a NEO-6M GPS. With a 18650 battery and a 0.96-inch white OLED, the frame mounts to the frame of a bicycle. The ESP32 provides an Access Point with a web server (see Figure 18). 

Once the mobile phone is connected to the web server, the user can report issues. A LoRa 868 Mhz connection is to be used to send the collected data to the FTP server located at the parish council, thus updating the citizens’ reported problems. The FTP server stores the data for later upload to the Esri Survey123^®^ application (see Figure 19).

Another proposed system, based on Survey123, allows for collecting information about street mobility issues. Once the survey has been opened, the user must select the means of transport that s/he uses daily to go to school. This question is related to analyzing the mobility used by the school’s students that the teachers responsible for the project intended to carry out. In the following steps, users should identify the constraint, take a photo, and register their GPS information using the available map. The second proposed system is based on an APP created in a STEAM context, reporting different types of issues, such as recycling, urban, or others (see Figure 20). Reporting can be done anonymously by default, or the user can choose to write his/her name. In the data screen, the user can view data on a map, share the data with other users using an MQTT Broker, or import the data shared by other users also using an MQTT Broker.

Air sensor setup: using the proposed system, the project consists of a sustainable, low-cost air quality monitoring setup (see Figure 21). The CO_2_ equipment was kept on the ceiling at the height of about 3 m above ground level during the sampling. Field data are gathered inside and outside the classroom and written every 4 min to the ThingSpeak and an MQTT broker. Data can be downloaded from a CSV file provided by the ThingSpeak channel, seen on the school website in a graphical view, or by subscribing to the MQTT channel. Figure 22 presents CO_2_ data collected on 2 December 2022, during the period in which the students were attending classes and PM2.5 and PM10 outside de school. The PM equipment was kept at the height of about 4 m above ground level during the sampling. Some spikes happened due to internet loss at school. In the afternoon, there was an increase in the number of students attending the classroom, intensified by the already saturated air, so the concentration of CO_2_ increased considerably in this period. 

Sound sensor setup: using the proposed system, the project consists of a sustainable, low-cost sound level meter setup (see Figure 23). During the sampling, the equipment was kept on the top of a table at a height about 1 m above ground level. Field data are gathered in a classroom and written to ThingSpeak every 30 s and to an MQTT broker. Data can be downloaded from a CSV file provided by the ThingSpeak channel, seen on the school website in a graphical view, or by subscribing to the MQTT channel. Figure 24 presents sound data collected on 12 January 2023, during the period in which the students were attending classes. 

Regarding Maths curricular integration, the kit was used in VET classes in general statistics domains and in organizing and interpreting statistical characters. Students engaged in activities giving answers to the following problem: “What sound intensity level is measured at different times of the day and/or at different locations in the school?” Students used a SmartSound level meter to collect data and existing data in CSV format from field 1 of channel number 1700097 of the ThingSpeak platform. Students were divided into pairs to answer a set of questions based on sound values recorded over the course of a day. Afterward, they answered a set of questions and recorded the observations/conclusions in the daily notebook.

Science students answered the problem question: “What is the level of sound intensity measured at different times of the day and/or at different locations at school?” To address it, students answered six contextualization questions regarding the problem under study. After that, students were subdivided into two work groups. The first one downloaded data available on the ThingSpeak platform and answered a questionnaire based on the available data. The second one received the SmartSound sonographer to collect data on a field trip inside the schoolyard. Data were collected from different locations in the school and recorded in a previously provided grid. Both groups answered two more questions based on the recorded observation.

SmartWaste sensor setup: using the proposed system, the project consists of a 20 kg aluminum load cell and an HX711 converter module with a 24 high-precision A/D converter chip monitoring setup (see Figure 25). The sensor equipment is to be kept inside de school kitchen. Data gathered by the sensor will be uploaded to the MQTT broker and available on the school website. The school website will be displayed at the canteen during meals. The operation of the equipment by the school canteen staff is intended to be very simple. They will only have to collect the waste from the students’ meals and insert it into the scale. Several buttons will be available, allowing data to be sent to the MQTT broker, reset the equipment, and calibrate it.

SmartGreen sensor setup: using the proposed system, the project consists of two kits, each one with its own sensors and microcontroller. Using IoT concepts and cloud storage, kits will derive vital crop information for performing agriculture activities. The system depicts the usage of embedded sensors in microcontrollers as it significantly enhances the potential performance of WSN technology and the system’s portability. Scalability is also important and will be addressed. Hence it will permit the addition of newer devices over the existing infrastructure without affecting the services and functional capabilities of the existing framework. Information on soil nutrient monitoring will be applied to predict crop health and production quality over time. The data report provided by the MQTT broker will allow decision-making regarding plants’ automatic irrigation. In addition to the automatic mode, the Smart Greenhouse application will provide a manual watering mode. A smart approach to watering plants means they get the exact amount of water needed. The drip irrigation system saves water by directly providing the correct amount of water to the roots. A sprinkler will be installed for watering over the top of higher plants. A water storage tank has a couple of water level sensors, also known as float switches for water tanks, that can sense or detect water level giving control over the existing water level in the storage tank. Controlling the water level in the storage tank is essential for keeping the system running efficiently. The system uses capacitive liquid-level sensing with an induction principle and high and low alarm options. MQTT push notifications will be sent to the app (see Figure 26) with real-time soil moisture, air humidity and temperature, and water level tracking.

### 4.2. Results

In this study, only data related to surveys applied to students will be highlighted. The remaining data have already been the subject of an in-depth evaluation of the authors’ unpublished manuscripts. 

In the 2019/2020 school year, and regarding students’ pre-test, from nearly 350 responses, 298 validated answers—students that responded to all questions—were considered eligible for the study. Pre-test data analysis revealed a lack of citizenship regarding students’ attitudes. When asked about their perceptions concerning their civic participation, having an active voice in society, and helping with troubleshooting, on a scale of agreement (totally disagree; partially disagree; neither agree nor disagree; partially agree; totally agree; no answer), their most frequent response was “neither agree nor disagree”. Students saw themselves as local community-engaged citizens. Still, when asked about their involvement in participatory citizenship activities in the last 12 months, only 25% of the respondents had some kind of participation. After applying strategies of participatory citizenship during the school year, from a population of nearly 600 students attending the school, 330 validated answers—students that responded to all questions—were considered eligible for the post-test study. Qualitative data was analyzed, regarding student surveys, with WebQDA software. The full study used a three main categories analysis, each with three subcategories. More detailed information about the analysis tree can be found in Santos et al. [12]. A huge increase in the number of students involved in participatory citizenship activities occurred during the school year. After implementing the strategies, 46% of students stated they were involved in implementing citizenship projects. Students’ major concerns were environmental problems (47.8%), being a better citizen (30%), social issues (6.7%), getting to know better the streets of my city (6.7%), mobility issues (2.2%), food issues (1.1%), and other issues (5.6%). Students involved in the activities implemented by the teachers during the school year tend to agree, and some of them also completely agree about their duties as participatory citizens. About being citizens who get involved in the community where they are integrated, only 14.6% of these students expressed negatively, contrasting with the 25.3% of the ones who did not implement the strategies. Globally, there was no change regarding students’ involvement in global issues. More information regarding phase 1, conducted in 2019, is available in Santos et al. [12].

The second DBR cycle was implemented in the 2021/2022 school year. Fifty-six students answered the pre-test, and 30 answered the post-test. Only validated answers—students that responded to all questions—were considered eligible for the study. The 26 missed answers in the post-test are related to COVID-19 students’ confinement and school dropout. The pre-test students’ perceptions show a positive impact regarding their participation. Students affirmed that they are “aware of the difficulties that our society is going through” and that “there is an increased desire to help others and evolve into a more united community”. Students also believe that it is paramount to help “families in need and support a kennel giving food”, and now they find themselves “capable of being more active in society”, so they should “exercise the right as a citizen to help”. School remains the place where they assume that they can have a voice. Students also think that they should engage in troubleshooting community problems. Nearly half of the students should help their community solve local problems. At a national level, only 40% feel the same way. Regarding the involvement in participatory citizenship activities during the 2020/2021 school year, 70% of the students have been involved in at least one activity. After implementing the citizenship strategies, students are more aware of their civic duty and the ability to be interventionists in their community. The number of students that were involved in participatory citizenship activities increased. The same happened regarding helping with issues in several domains. 

Regarding the type of volunteer involvement, CS could be classified as contributory, collaborative, or co-created. In contributory involvement, citizens merely contribute to data collection and eventually help analyze or disseminate the data. On the other hand, a collaborative approach will allow citizens to analyze samples and data and, sometimes, even help design the study, interpret the gathered data, draw conclusions based on the results and disseminate them. Finally, the co-creation approach allows citizens to participate at all stages of the projects. They define the questions, draw hypotheses, discuss the results, and answer questions. The EU-Citizen.Science (https://eu-citizen.science/) (accessed on 7 January 2023) is a European platform for sharing CS projects. Many CS projects are registered in Portuguese and from other European countries, aiming to solve a local or national problem. The seven Portuguese projects are Explorator, GelAvista, Invasoras, CoAstro, BioRegisto, MosquitoWeb, and GripeNet. They are all framed in the contributory volunteer involvement field. Regarding international projects, Crowd4SDG, Achieving a new European Energy Awareness (AURORA), Butterfly Migration, Hush City, or Science in the city, all, as with the Portuguese projects, citizens are merely contributing to science. The CitiMeasure project (https://citimeasure.eu/) (accessed on 7 January 2023) aims to raise awareness about citizens’ importance in implementing citizen science initiatives in their local community. Focused on a co-creation process, the project is grounded in three main instruments: comparability of air quality measurement initiatives, digital inclusion competencies, and citizen science initiatives’ influence on behavior and policy changes. CitiAIR is a tool provided by the project that brings together European projects related to air quality. The SmartAir (https://citimeasure.eu/initiatives/smartair/) (accessed on 7 January 2023) project is also referenced here with the main missions of your measure/monitoring air quality, promoting smart/sustainable mobility, and raising awareness/education on the issue of air quality. Certainly, one of the main goals of the projects is to improve citizens’ quality of life and, eventually, to increase their scientific literacy. However, they do not allow effective citizen participation in local decision-making, just as not being co-created by citizens; they do not allow for an increase in the level of citizens’ co-responsibility. By going through all the phases of the projects, the students became more aware of communal problems as they gained a more comprehensive knowledge of them. Not only did they recognize the existence of the problems, but they also discussed and found together with teachers and stakeholders the best ways to mitigate them. Students also proceeded to analyze the collected data and disseminated the results among peers and even at the community level.

The construction of collaborative projects between the educational agents, starting with the identification of issues and the search for possible solutions, going through the implementation of the action plan, and ending with sharing the results with the local community, were catalyzers that fostered students’ participatory citizenship. Indeed, CS strategies aided by collective awareness platforms, sensors, and micro-controllers, engaging local society in communal problems, can contribute to a more effective dialogue between the educational agents and promote students’ participatory citizenship.

### 4.3. Study Limitations

The data provided by the CO_2_ sensor outside the school did not undergo significant changes in their readings, not meeting what was expected to happen. The equipment was eventually reconverted for internal reading of the CO_2_ level inside the classrooms after the breakout of the COVID-19 pandemic situation. Another limitation of the study is related to data communication based on LoRa technology, which was not carried out due to the inexistence of LoRa Gateways in the target city of the study. Not being able to take advantage of LoRa long-distance communication, researchers opted for Wi-Fi and MQTT protocol to upload data. Built on top of the TCP/IP, MQTT is a standard for IoT communications, with low bandwidth consumption, efficient distribution of information, and low power consumption.

IoT technologies have some seriously limited resources that need to be addressed, such as energy, CPU power, and memory. The most challenging one was energy consumption. Data routing and transfer are of great importance; in wireless communications, it consumes more energy than processing. SmartGreen and the SmartAir equipment have built-in rechargeable batteries that will be recharged using a solar panel that should be installed later this school year. By default, the ESP32 module has built-in 4 MB flash memory and may also be ordered with a custom flash size of 8 or 16 MB. The data will be saved in the flash memory, even when the ESP32 resets or power is removed. Wi-Fi credentials and variables state, or any other type of data that needs to be saved permanently, will be saved in the flash memory. The flash memory reading process is infinite, but the writing process can only be done, in most devices, for about 100,000–1,000,000 operations.

## 5. Conclusions

This study reports on CS strategies developed at the curricular level, integrating communal environmental issues in school projects supported by the Internet of Things, aiming to foster students’ participatory citizenship, promote the dialogue and communication between local actors, and contribute to informing municipal education policies.

Students’ prior lack of citizenship and engagement in community issues was addressed by new pedagogies that contributed to developing autonomous, responsible, and solidary citizens, focusing on communal issues like climate change. Several documents recently published by the Portuguese Ministry of Education reveal concerns about the lack of work done in the national public school about citizenship education. The Portuguese Citizenship Education Guidelines, the National Education Strategy for Citizenship, and the Sustainability for Environmental Education Referential are examples of this concern. Moreover, interdisciplinarity pedagogical practices should include real-time data-gathered activities with subsequent classroom analysis and interpretation. The study’s findings suggest that to build a bridge for participatory citizenship, teachers should engage students in communal environmental issues in a STEAM approach or under Domains of Curricular Autonomy activities. CS strategies aided by collective awareness platforms, sensors, and microcontrollers were catalyzers that fostered students’ participatory citizenship. There was a huge involvement of the parish council and municipality in the training sessions and in the educational community awareness activities, contributing to a greater articulation and concertation of agendas between the school, municipality, and parish council. Several projects were created under the aegis of the Smart School Lab, which substantially improved the quality and sustainability of educational intervention projects.

The study was carried out in a secondary school, so future work should target basic education students starting in the first cycle of basic education. Regarding the IoT concept, data privacy is always important, considering information published online [114]. Some sensitive data, such as the possibility of accessing the users’ location, should be further studied, including better data encryption. In addition to the already mentioned issues, uninterrupted network connectivity, security-related issues, and the physical safety of objects are also a concern.

## Figures and Tables

**Figure 1 sensors-23-03070-f001:**
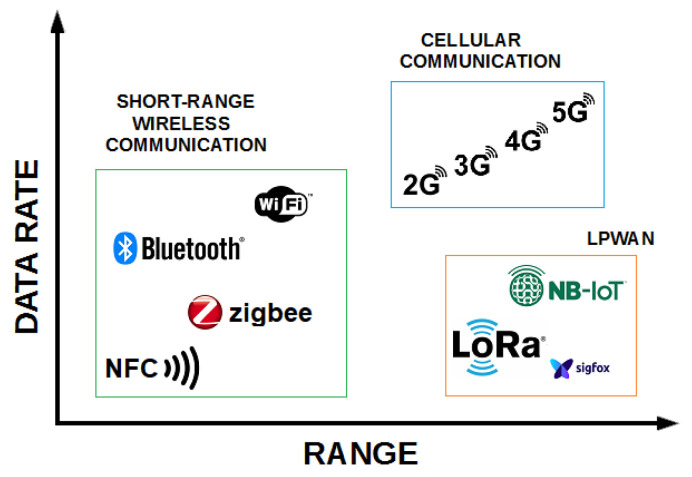
Data rate vs. a range of several IoT communication protocols.

**Figure 2 sensors-23-03070-f002:**
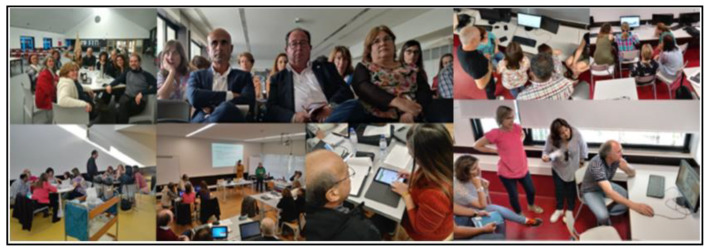
Smart School Workshops.

**Figure 3 sensors-23-03070-f003:**
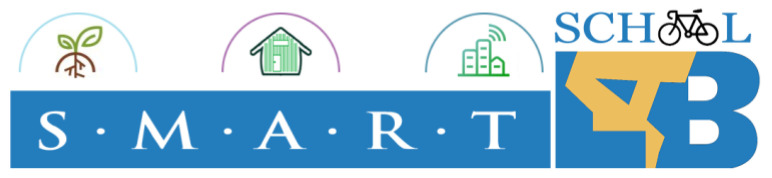
Smart School Lab logo.

**Figure 4 sensors-23-03070-f004:**
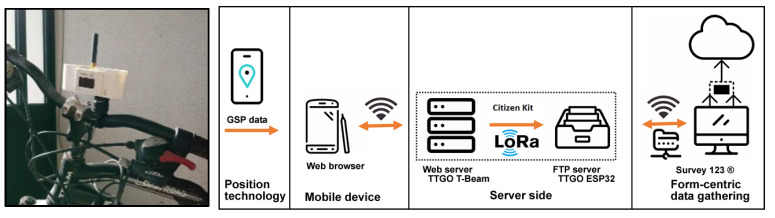
First proposed system architecture.

**Figure 5 sensors-23-03070-f005:**
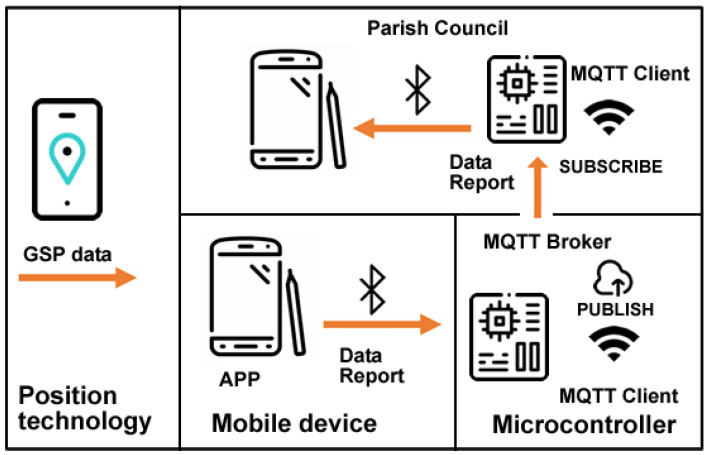
Second proposed system architecture.

**Figure 6 sensors-23-03070-f006:**
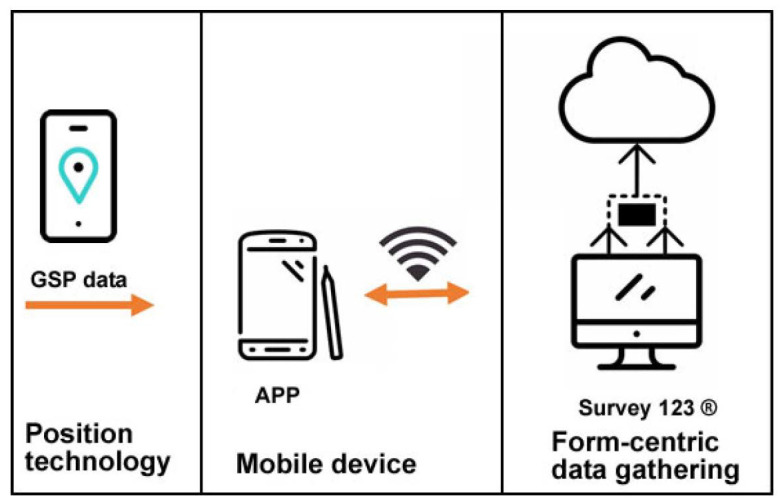
Third through sixth proposed system architecture.

**Figure 7 sensors-23-03070-f007:**
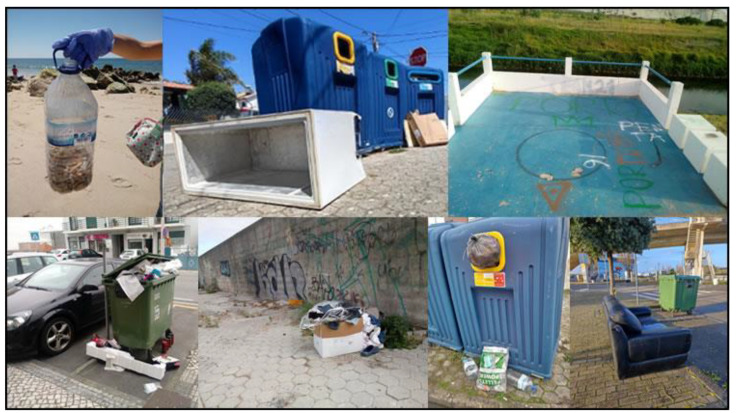
Garbage, recycling, and other issues in the local community.

**Figure 8 sensors-23-03070-f008:**
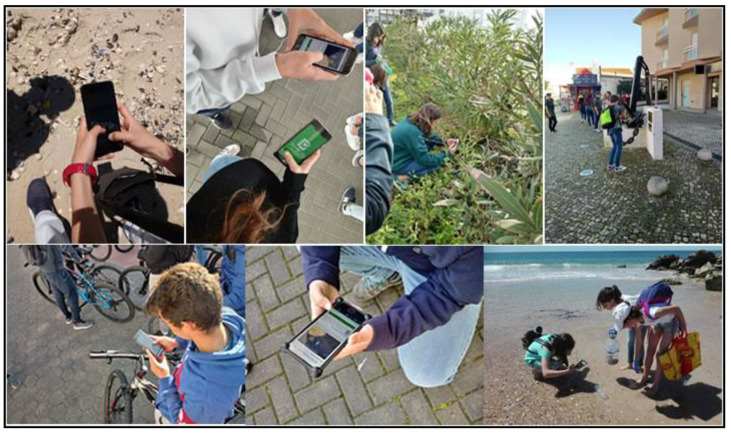
Students gathering data with Survey123.

**Figure 9 sensors-23-03070-f009:**
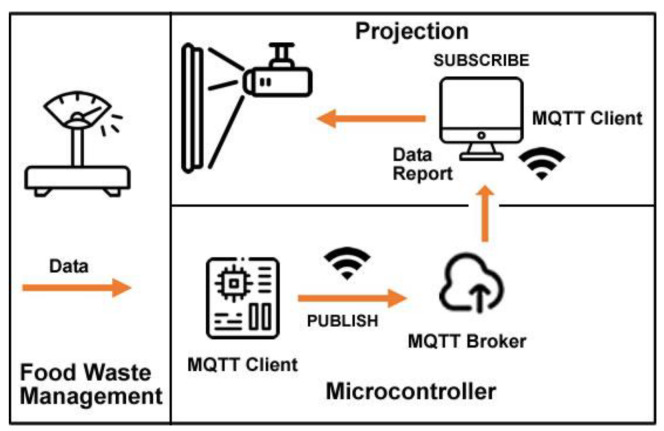
SmartWaste System Architecture.

**Figure 10 sensors-23-03070-f010:**
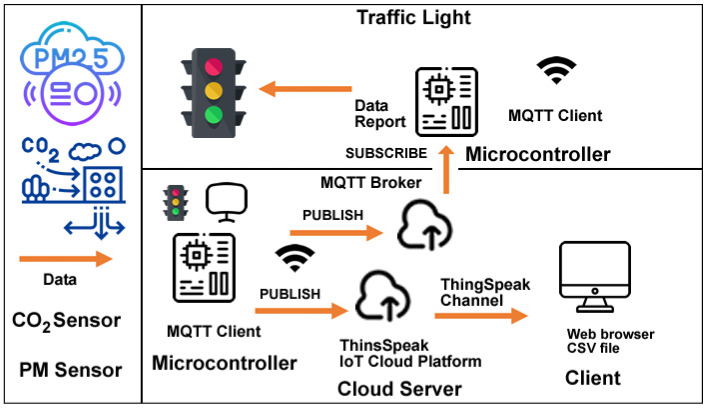
SmartAir System Architecture.

**Figure 11 sensors-23-03070-f011:**
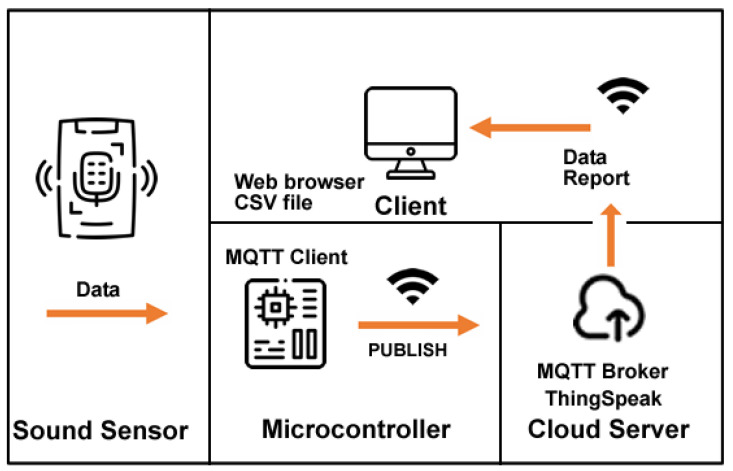
SmartSound System Architecture.

**Figure 12 sensors-23-03070-f012:**
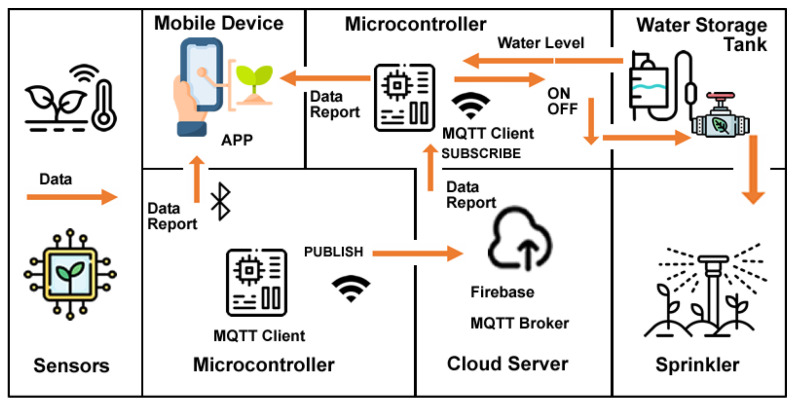
SmartGreen System Architecture.

**Figure 13 sensors-23-03070-f013:**
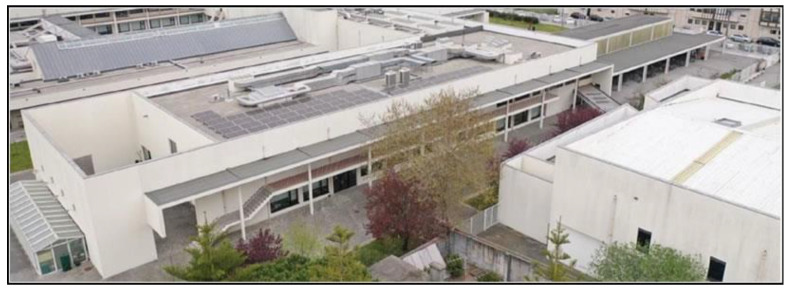
Solar panels installed on the roof of the target-school.

**Figure 14 sensors-23-03070-f014:**
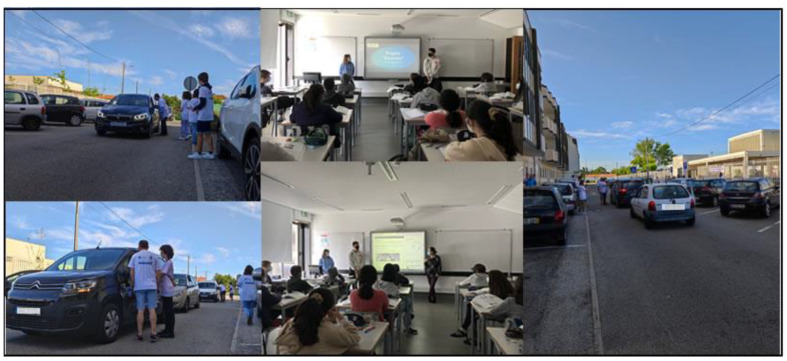
SmartAir community awareness sessions.

**Figure 15 sensors-23-03070-f015:**
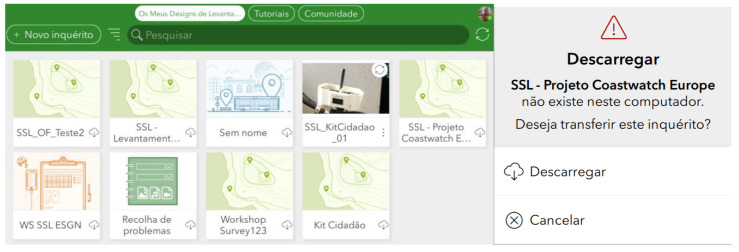
Survey123 APP.

**Figure 16 sensors-23-03070-f016:**
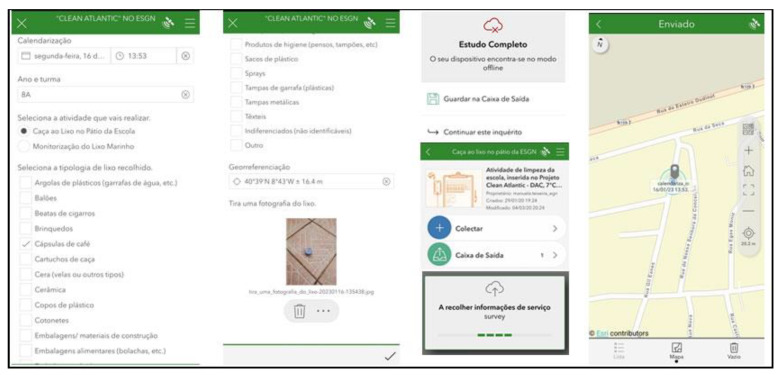
Collecting data with Survey123.

**Figure 17 sensors-23-03070-f017:**
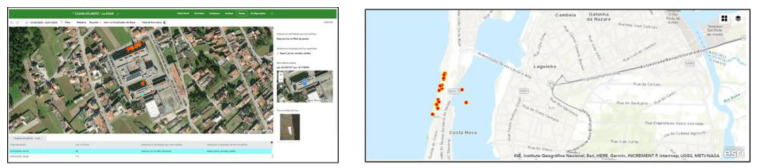
Survey123 map with a detailed view of collected data. Red dots correspond to the GPS location of each collected waste.

**Figure 18 sensors-23-03070-f018:**
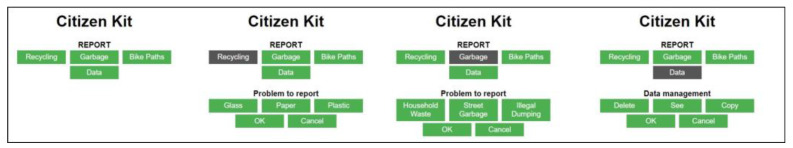
Web server HTML pages.

**Figure 19 sensors-23-03070-f019:**
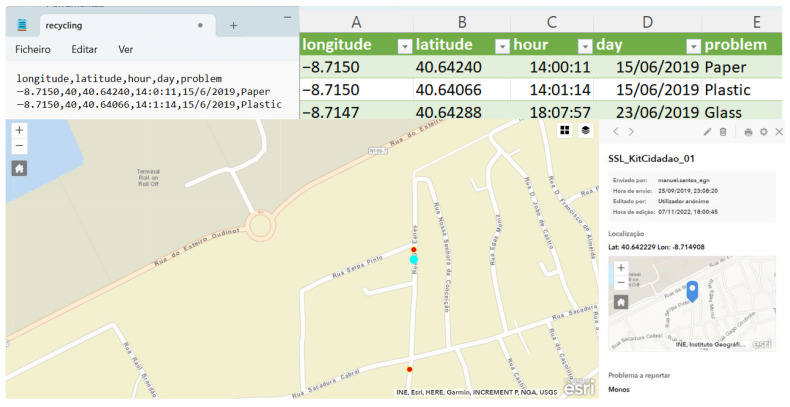
FTP server and Survey123 map with reported issues. Red and blue dots correspond to the GPS location of each collected waste.

**Figure 20 sensors-23-03070-f020:**
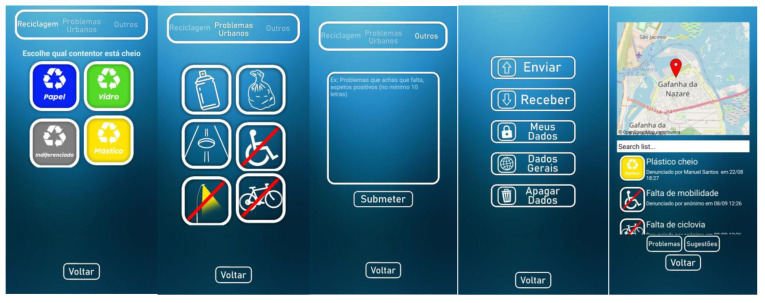
BikeTrack app.

**Figure 21 sensors-23-03070-f021:**
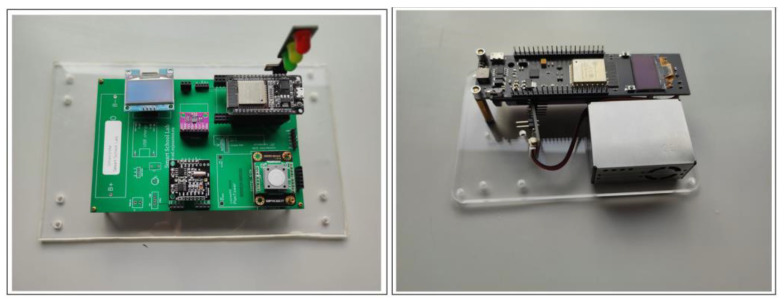
SmartAir kits.

**Figure 22 sensors-23-03070-f022:**
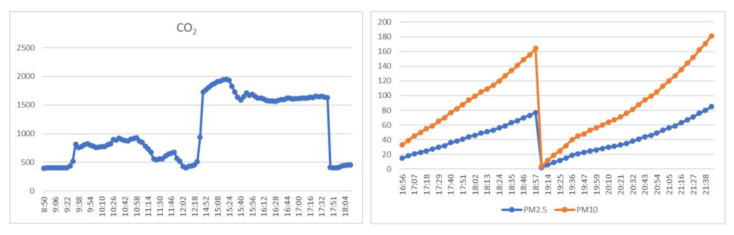
One day CO_2_ level inside a classroom and PM outside the school.

**Figure 23 sensors-23-03070-f023:**
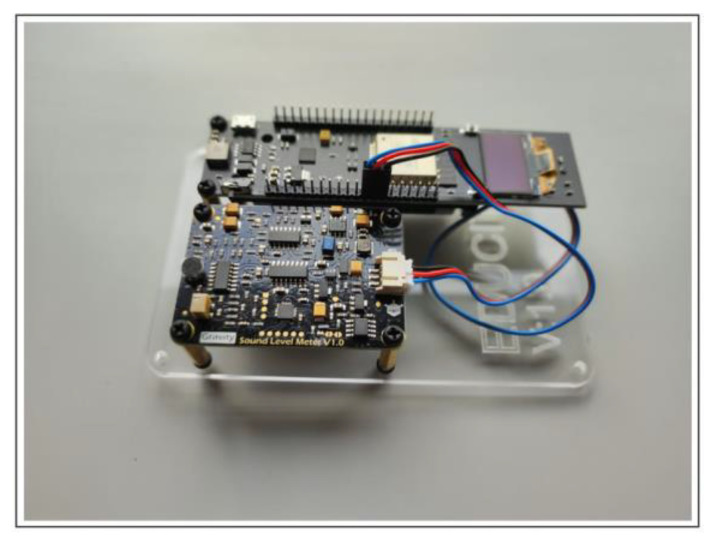
SmartSound kit.

**Figure 24 sensors-23-03070-f024:**
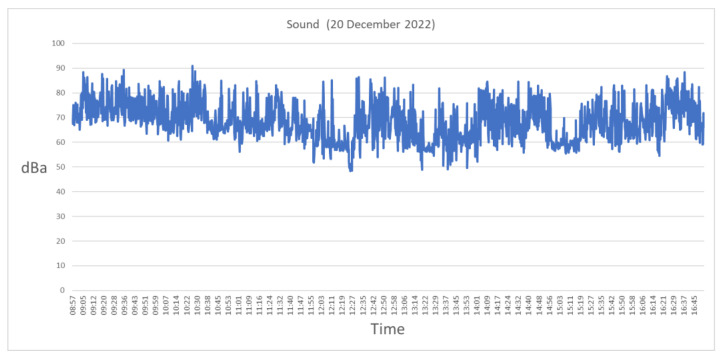
One day sound level inside a classroom.

**Figure 25 sensors-23-03070-f025:**
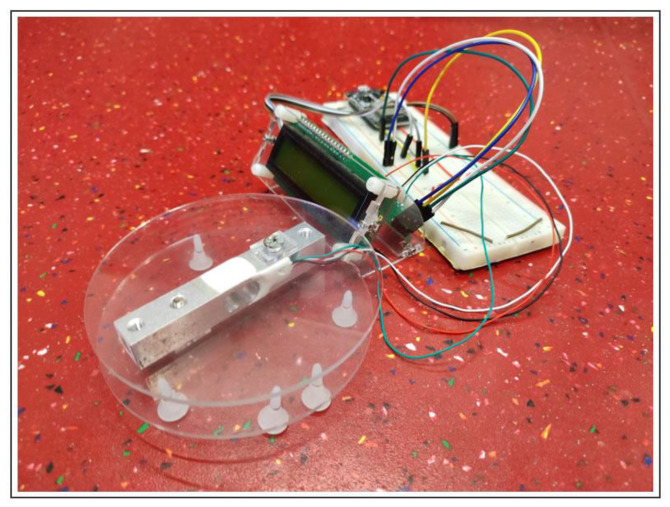
SmartWaste kit.

**Figure 26 sensors-23-03070-f026:**
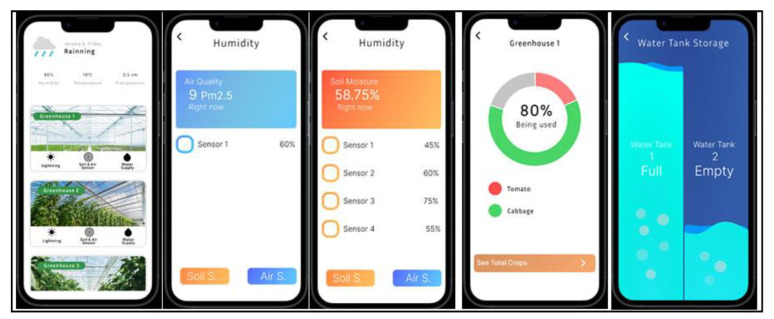
SmartGreen APP.

**Table 1 sensors-23-03070-t001:** Assessment, teaching, and learning strategies and measurement indicators.

Action Strategies	Measurement Indicators
Encouraging students and others to participate in active citizenship activities/projects.	The number of active citizenship activities/projects. The number of participants (teaching and non-teaching staff, parents and guardians, and students).
Digital Academy for Parents (Promotion of basic digital skills).	The number of participating Parents.
Reinforcement of clubs with an artistic aspect in the school community.	The number of activities performed.
Increase in partnerships with local entities linked to culture.	The number of partnerships.
Promotion of the aesthetic-artistic development of the community through experimentation, fruition, and critical intervention.	The number of participants. Congruence between the AEGN citizenship education strategy plan and the final report.
Awareness of financial literacy.	Projects/activities carried out.
Involvement of students in the Participatory Budget of the School.	The number of students involved.

**Table 2 sensors-23-03070-t002:** Create opportunities for student participation in decisions about their education strategies and measurement indicators.

Action Strategies	Measurement Indicators
Sensitization of teaching and non-teaching staff to changes and initiatives regarding student participation in decision-making processes.	At least one class assembly in the first cycle per semester and at least three in the second and third cycles and secondary education.
Students define the code of conduct to be posted in the classroom.	At least two assemblies of delegates and sub-delegates from the second and third cycles, and secondary education, concretizing aspects related to the “Students’ Voice”.
Development of spaces for the democratic participation of students in the Group.	The number of debates in the classroom.
Promotion of actions that mobilize students for active citizenship in the community.	The number of students participating in the Youth Parliament.
Definition of lines of action that enhance the hearing of students’ voices.	Satisfaction degree.

**Table 3 sensors-23-03070-t003:** Sustainability, health and well-being strategies and measurement indicators.

Action Strategies	Measurement Indicators
Identification of potentialities and constraints in the ecological behavior of the school population.	Environmental audit.
Holding “Meetings with…” specialists in the environmental area, which promote lifestyle changes and sustainable development in the educational community.	Annually, at least one meeting with all elements of the educational community and/or with a selected target audience.
Promotion of initiatives that lead the school community to reduce/reuse/recycle.	The number of initiatives.

**Table 4 sensors-23-03070-t004:** Study’ specific objectives and related actions.

Objective	Actions
Promote students’ involvement at school and in the community	Use older students as mentors Contact with real work contexts. Diversification of teaching and learning strategies. Use strategies closer to students’ daily lives.
Promote the dialogue and communication between local actors	Improve the quality and sustainability of educational intervention projects.
Contribute to informing municipal education policies	Greater articulation and concertation of agendas between school, municipality, and the parish council.

**Table 5 sensors-23-03070-t005:** Teachers’ and stakeholders’ training sessions and workshops on civic co-creation.

Workshop	Teachers	Stakeholders
Smart Schools: the creation of georeferenced digital narratives with StoryMaps^®^	19	3
Smart Schools: sharing of georeferenced information through the Geonets social network^®^	18	1
Smart Schools: a network of geomentors in civic co-creation	10	0
Smart Schools: creating smart georeferenced forms with Survey123^®^.	13	0

**Table 6 sensors-23-03070-t006:** Sensors in smart environments (adapted from [99]).

Properties	Measurement
Physical properties	Pressure, temperature, humidity, flow, gas, moist, light
Motion properties	Position, velocity, acceleration
Contact properties	Strain, force, torque, slip, vibration
Presence	Tactile/contact, proximity, distance/range, motion
Identification	Personal features, personal ID

**Table 7 sensors-23-03070-t007:** Citizen sciences strategies.

Goals	Strategies	Answers	Users
Citizens’ engagement in collecting data	Waste/garbage Recycle Bike paths/safety routes Toponymy Charity	899	235 users
Sensors collecting data	Air (PM + CO_2_) Sound Greenhouse	2862 + 8790 49,584 N/A ^1^	N/A ^1^

^1^ N/A—not applicable.

**Table 8 sensors-23-03070-t008:** Citizen sciences strategies—citizens’ engagement in data collection.

Strategies/Scope	Participants	Study Area
Waste/garbage (DAC)	Design: a first project regarding sea waste was designed by two teachers and four students, and a second one, regarding street and school ground waste, was designed by 4 teachers and nearly 10 students. Data collection: Regarding the first project, three environmentalists, three teachers, 19 students, and two researchers from the University of Aveiro participated in the collection of sea waste. As for the second project, four teachers and more than 40 students participated in the garbage collection at the school.	Data collection takes place in several places: on the city’s streets, the municipality’s beaches, and on the school grounds of the target school of the study.
Recycle (STEAM)	Design: the first project was designed by the researcher and a couple of students. The second project was designed in a STEAM context by two teachers (Art and Technology) and more than 15 students. Data collection: all citizens will be able to collect data with BikeTrack app.	Data collection will take place on the streets of the city of the target school of the study.
Bike paths/safety routes (DAC/STEAM)	Design: the first project was designed by four teachers during a teacher training action in co-creation with the remaining 15 teachers who attended the training action and external stakeholders, among others, the Parents’ Association. The second was described in the previous row of this table (Recycle). Data collection: in the first project, more than 50 students collected data and produced a digital narrative using StoryMaps. Regarding the second one, all citizens will be able to collect data with the BikeTrack app.	The data collection took place on the streets of the city of the target school of the study.
Toponymy (DAC)	Design: the project was designed by three teachers during a teacher training action in co-creation with the remaining 15 teachers who attended the training action and external stakeholders, among others, the Parents’ Association. Data collection: more than 20 students collected data and produced digital narratives using StoryMaps.	The data collection took place on the streets of the city of the target school of the study.
Charity (DAC)	Design: more than 20 students and one teacher were involved in the design and development of the system. Data collection: besides the aforementioned participants, all volunteers from Charity at the local community level were involved in the project collecting data.	The project was carried out in the community of the target school of the study.

**Table 9 sensors-23-03070-t009:** Citizen sciences strategies—use of sensors to collect data.

Strategies/Scope	Participants	Study Area
Air (DAC)	Design: 10 students and two teachers were involved in the design and development of the system. Awareness: more than 50 students, the Parent Association of the target school, the parish council, four researchers from the Department of Environment and Planning of the University of Aveiro, and local environmental entrepreneurs.	The data was collected in two classrooms and at the study’s target school’s entrance. The awareness sessions were carried out nearby and inside the school.
Sound (STEAM)	Design: 22 students and four teachers were involved in the design and development of the system. Data collection: more than 50 students collected data and conducted mathematics and Physics curricular activities.	The data collection was carried out in several classrooms and other places inside and outside the target-school of the study.
Waste (DAC)	Design: four students are designing and developing the project with the support of three teachers. The students conceived the project based on a challenge launched by the school board.	The project will be carried out inside the target school of the study.
Greenhouse (DAC)	Design: three students and five teachers are participating in the development of the system. The students conceived the project based on the challenge launched by the teachers responsible for the Living Science school project.	The project will be carried out inside a greenhouse in the target school of the study.

## Data Availability

The datasets generated and/or analyzed during the current study are not publicly available due to individual privacy but are available from the corresponding author upon reasonable request.

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
