# Peer review of "Building the Bridge to a Participatory Citizenship: Curricular Integration of Communal Environmental Issues in School Projects Supported by the Internet of Things"

_sensors, 2023, doi:10.3390/s23063070_

Round 1
Reviewer 1 Report
In this paper authors report their experience using citizen science techniques used in school projects.
The introduction is too succint (compared to the overall length of the paper). Also, it fails to clearly state the overall purpose of this contribution.
Section 2 covers a lot of topics (i.e., citizenship education, IoT and Smart Cities, IoT Communication) Technologies, and it is difficult for the reader to interconnect them with the purpose of the paper. Also, section 2.3 is way too technical (Figures 2-4 and Table 1) for the overall aim of the paper.
Authors mention that they have collected several amounts of data in their experimentation, but little details are provided on this; how much data? Where are data stored?
Overall, I have really liked the amount of details that authors have provided so their study is replicable to other domains. However, I think that the paper is too dense for the reader in its current form. Hence, I suggest extending the Introduction, and reducing section 2 (and even 3).
Author Response
Dear reviewer,
Thank you for your very careful review of our paper, for the corrections, the comments, and the suggestions. To consider all of them, a major revision of the paper has been carried out, so we believe the paper has been significantly improved. Please see the attachment.

Reviewer 2 Report
The paper presents a study of citizen science strategies exploiting IoT inside school projects. The final goal is to foster students’ participatory citizenship and promote dialogue and communication between local actors.
The paper's topic is interesting and in line with the special issue. However, it is not easy to follow the process and understand the real contribution of this study. It is clear the gap that the study wants to address, but the discussion is too brief to really understand the contribution. As a matter of fact, the paper just presents the different apps (and their working principles) and scenarios, without highlighting a final evaluation with the community to prove that the goal mentioned above is reached. Moreover, as systems similar to the ones developed already existed, a discussion considering also these systems should be presented: what are the differences? what advantages has co-creation brought?
Some minor concerns: there are a lot of typos (also in the abstract), some figures are not easy to read (e.g., figure 23), formulas are in form of figures and the equation's terms are not always explained.
The paper has potential but needs to be better refined: the study's results should be better explained and the discussion improved. Even in the introduction, the real contribution is not clear.
Author Response

(The authors gave the same response as above.)

Reviewer 3 Report
Dear authors, I enjoyed the manuscript and learned many things from it. You wrote the manuscript clearly and perfectly. Congratulations! Still, I have some criticism and questions.
You should give information about citizenship types. There is not enough information about them. There are not enough information about these: "1) State- 420 of-the-art literature review; 2) Document analysis; 3) Geomentors mobilization; 4) Local 421 needs and contextual purposes, regarding School and Municipality; 5) Roles and commitments". I need more clarification about that.
Your results are priceless, but you should discuss them with literature studies, especially citizenship studies.
The results and conclusion sections are written clearly.
Study limitations are very curial. Some studies ignore this explanation.
Author Response

(The authors gave the same response as above.)

Reviewer 4 Report
The submitted paper presents an intriguing topic and has a lot of potential as a scientific paper. However, it appears to be more of a project report rather than a scientific paper. As a kind of project report it describes generically the activities, approach, methodology, and results of the various parts, while a scientific paper is a formal publication that presents original research, normally including a hypothesis, methodology, results, and conclusion. In order to transform the project report into a scientific paper, the author may need to include more in-depth analysis of the results and a discussion of the implications of the findings for the field of study.
I have no problems with how the paper is formatted and written; it adheres to the conventions of scientific writing, such as the use of proper citation and referencing.
Author Response

(The authors gave the same response as above.)

Round 2
Reviewer 1 Report
Authors have addressed all the issues I suggested in my previous review.
Author Response
Dear Reviewer,
We want to express our gratitude for taking the time to review our article and for providing us with valuable comments and suggestions. We appreciate the effort and thoughtfulness you put into your feedback.
Sincerely,
Manuel Santos
Reviewer 2 Report
I really appreciated the work made by the author to improve the manuscript and now I think the paper is ready to be accepted.
Author Response

(The authors gave the same response as above.)

Reviewer 4 Report
The integration of communal environmental issues in school projects can be a powerful tool for fostering participatory citizenship among students. Such projects can be made more effective by leveraging modern technology, such as the Internet of Things (IoT), to gather and analyze environmental data. This paper explores the role of IoT in supporting the curricular integration of communal environmental issues in school projects, with a focus on its potential to enhance students' understanding of and engagement with environmental issues.
The paper has some interesting ideas about the potential of IoT (this part of the paper is highly irrelevant) to support the curricular integration of communal environmental issues in school projects. However, it suffers from a number of limitations. The paper lacks sufficient detail, and it provides only a superficial description of the results. The paper also does not provide adequately specific recommendations or discuss potential challenges. As a result, it is not clear what relevant findings can be drawn from the study, and it is difficult to assess the scientific value of the paper.
Author Response
Dear Reviewer,
The paper discusses several potential challenges, such as: (1) ethical and legal challenges; (2) technical challenges; (3) engagement and participation challenges; and (4) social challenges.
(1) Ethical and legal challenges
One potential challenge of using citizen science strategies with students is ensuring that all ethical and legal requirements are met. The authors took into consideration that all ethical and legal requirements were met. This included obtaining informed consent, ensuring data privacy and security, and complying with any relevant regulations or laws, so that the approach to citizen science was responsible and sustainable.
(2) Technical challenges
Implementing citizen science projects using the Internet of Things and a network of users may require technical expertise that is not readily available to all participants. Several co-creation workshops were held, allowing the educational community to acquire technical skills. The paper also discusses authors’ experiences with implementing LoRa technology. Despite authors’ initial excitement for LoRa's long-range capabilities, we encountered technical difficulties that hindered our progress. As a result, we had to pivot and turn to the MQTT protocol to achieve the results we were looking for. While this was a challenging setback, it ultimately led us to explore new solutions and expand our knowledge in the field.
(3) Engagement and participation challenges
Another potential challenge of using citizen science strategies with students is ensuring that all participants remain engaged and motivated throughout the project. Engaging and motivating participants to actively participate in the project were a major challenge, and it was crucial to address these challenges to ensure the success of the project and yield meaningful data. Some potential engagement and participation challenges were discussed such as students’ lack of interest. To overcome these challenges, authors clearly communicated the project's objectives and potential benefits to participants, provided incentives for participation, and create a user-friendly and accessible platform/technology for data collection. Additionally, trust with participants was built, addressing any concerns they may have, increasing engagement and participation. By taking these steps, the study could achieve its goals and provided valuable insights for researchers and stakeholders alike.
(4) Social challenges
Citizen science projects are often community-driven and require collaboration and cooperation between different groups of people. Social challenges were addressed, and guidance on how to overcome them were provided.
We also would like to provide some justification for the recommendations presented in the conclusion section, specifically regarding the importance of interdisciplinary pedagogical practices and engaging students in communal environmental issues.
Our study found that real-time data gathered activities with subsequent classroom analysis and interpretation can be an effective way to promote interdisciplinarity. By integrating data from multiple disciplines, students can develop a more comprehensive understanding of complex issues, such as environmental concerns.
Moreover, we believe that engaging students in communal environmental issues is crucial for building participatory citizenship skills. Through a STEAM approach or Domains of Curricular Autonomy activities, students can apply their knowledge and skills to real-world problems and become active participants in their communities.
Overall, our recommendations aim to promote a more integrated and socially responsible approach to education, where students are equipped with the skills and knowledge necessary to address complex issues and contribute positively to society.
Sincerely,
Manuel Santos